# VoiceMixer: Adversarial Voice Style Mixup

**Sang-Hoon Lee**[1]  **Ji-Hoon Kim**[2]  **Hyunseung Chung**[2]

**Seong-Whan Lee**[2*]

{sh_lee, jihoon_kim, hs_chung, sw.lee}@korea.ac.kr
[1]Department of Brain and Cognitive Engineering, Korea University, Seoul, Korea
[2]Department of Artificial Intelligence, Korea University, Seoul, Korea

## Abstract

Although recent advances in voice conversion have shown significant improvement, there still remains a gap between the converted voice and target voice. A key factor that maintains this gap is the insufficient decomposition of content and voice style from the source speech. This insufficiency leads to the converted speech containing source speech style or losing source speech content. In this paper, we present VoiceMixer which can effectively decompose and transfer voice style through a novel information bottleneck and adversarial feedback. With self-supervised representation learning, the proposed information bottleneck can decompose the content and style with only a small loss of content information. Also, for adversarial feedback of each information, the discriminator is decomposed into content and style discriminator with self-supervision, which enable our model to achieve better generalization to the voice style of the converted speech. The experimental results show the superiority of our model in disentanglement and transfer performance, and improve audio quality by preserving content information.

## 1 Introduction

Voice conversion (VC) is the task of transferring the target voice style to the source speech while keeping the content information of the source speech. VC is also called voice style transfer (VST), and it shares a long history with the objective to clone someone's voice. There is even a potential risk of usage in crime such as a voice spoofing (Kinnunen et al., 2012), and also in various applications in entertainment (Nachmani and Wolf, 2019), education (Sisman et al., 2020), security (Wu and Li, 2016), and voice restoring (Yamagishi et al., 2012). Although deep learning made the breakthrough in the VC domain, there still remains challenging problems for real-world application such as low audio quality or similarity to target voice style.

Usually, traditional VC systems require the same utterances for different speakers to train properly. However, it is hard to collect such parallel data for many speakers, and extension to many-to-many VC systems becomes a laborious task. To overcome this problem, several methods have been developed. First, generative adversarial networks (GAN) based models (Kaneko and Kameoka, 2018; Kaneko et al., 2019, 2020; Kameoka et al., 2018) use adversarial feedback with cycle-consistent loss to train with non-parallel data. However, it is hard to train these models, and they produce lacking audio quality and transfer performance. The flow-based VC model, Blow (Serrà et al., 2019), is also a non-parallel VC model with normalizing flows using the hyperconditioning mechanism.

Despite effort to transfer the voice style in non-parallel settings, these models are not able to sufficiently disentangle content and style from the source speech, and thus the converted speech

---

*Corresponding author

35th Conference on Neural Information Processing Systems (NeurIPS 2021).

still contains the style of source speech. To overcome this limitation, AUTOVC (Qian et al., 2019) utilizes a simple autoencoder. The carefully designed fixed-length based information bottleneck disentangles the content and style information. For better disentanglement, IDE-VC (Yuan et al., 2021) followed the AUTOVC framework with information-theoretic guidance. AdaIN-VC (Chou et al., 2019) and AGAIN-VC (Chen et al., 2020) employs the instance normalization (Ulyanov et al., 2016) to remove the global style information. Additionally, AGAIN-VC makes use of the activation function as an information bottleneck with a small size of content embedding. However, these models have a trade-off between the audio quality and the disentanglement performance. In the process of disentanglement, the loss of content information results in low audio quality with missing linguistic information. Also, they have to find the proper size of information bottleneck heuristically.

Text transcriptions can be used to guide content embedding to learn only linguistic information (Biadsy et al., 2019; Zhang et al., 2019). These models have to be jointly trained with the text-to-speech (TTS) model to encode the linguistic information based on the attention alignment from autoregressive TTS system (Shen et al., 2018). However, they require text transcriptions for training.

Recently, self-supervised representation learning is adopted to extract important representation in speech representation learning task (Oord et al., 2018; Wang et al., 2020a). Predicting the future latent representation can make the model learn useful information without labeled data. However, such self-supervised representation learning has not yet gotten the attention in voice conversion task.

In this paper, we present VoiceMixer, which can decompose and transfer voice style through a novel similarity-based information bottleneck and adversarial feedback. We introduce self-supervised representation learning to disentangle and transfer voice style without any text transcription and additional information extracted from the external feature extractor. Self-supervised similarity based information bottleneck disentangles the content and style without effort to find the proper downsampling size. Also, we propose an adversarial voice style mixup to learn the latent representation of the converted speech. We first disentangle the discriminator into content and style discriminator. The hidden representations of generator guide each discriminator as conditional information. Through adversarial feedback of disentangled discriminators, the generator has better generalization on the converted speech. The main contributions are as follows:

- We propose the similarity-based information bottleneck with self-supervised representation learning, which can disentangle content and style with only a small loss of content information. This preservation improves the audio quality of converted speech compared to previous methods.
- For better generalization of the converted speech, we propose an adversarial voice style mixup, which learns the converted speech by adversarial feedback with self-supervised guidance, even though the converted speech does not have ground-truth audio.
- Through various subjective and objective evaluations, we demonstrate that VoiceMixer has better disentanglement and transfer performance than other baselines in both many-to-many and zero-shot voice style transfer scenarios on the real-world VCTK dataset.

## 2 Background

AUTOVC disentangles content and style information from the source speech, and transfers the voice style of target speech through information bottleneck (Qian et al., 2019). The simple autoencoder framework of AUTOVC consists of three modules; a speaker encoder $f_s(\cdot)$, content encoder $f_c(\cdot)$, and a decoder $g(\cdot, \cdot)$. During training, this model only requires self-reconstruction with a fixed-length information bottleneck to disentangle the content and style information.

$$\boldsymbol{S}_1 = f_s(\boldsymbol{X}_{1,A}), \ \boldsymbol{C}_A = f_c(\boldsymbol{X}_{1,A}), \ \hat{\boldsymbol{X}}_{1 \to 1, A} = g(\boldsymbol{S}_1, \boldsymbol{C}_A) \tag{1}$$

Here, $\boldsymbol{X}_{1,A}$ refers to the utterance "$A$" from the source speaker "1". $\boldsymbol{S}_1$ denotes speaker information in the speaker "1", $\boldsymbol{C}_A$ denotes content information of the utterance "$A$", and $\hat{X}_{1 \to 1, A}$ is self-reconstructed speech which contains the content information $\boldsymbol{C}_A$ and matches the speaker characteristics $\boldsymbol{S}_1$. Although it is a very simple way to decompose each information, a proper information bottleneck size $\tau$ is necessary. Formally, this can be represented as follows:

$$\boldsymbol{H}(:, \lfloor t/\tau \rfloor) = \boldsymbol{C}_A(:, t) \tag{2}$$

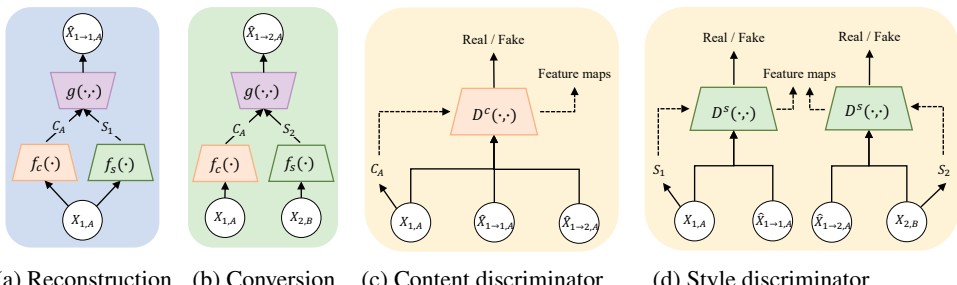

(a) Reconstruction  (b) Conversion  (c) Content discriminator  (d) Style discriminator

Figure 1: Overall framework of VoiceMixer.

where $H$ denotes the downsampled feature for time indices $t \in \{1, \cdots T\}$. When the fixed-length information bottleneck size $\tau$ is "too narrow", the model has higher reconstruction quality but has poor voice style transfer performance. On the other hand, when the $\tau$ is "too wide", the model has higher voice style transfer performance but has lower reconstruction quality. In the process of separating content and style information, some content information is lost even with proper bottleneck size. Therefore, missing some content information in converted voice is inevitable.

## 3 VoiceMixer

In this paper, we propose a similarity-based information bottleneck by self-supervised representation learning. For adversarial feedback, we disentangle the discriminator to train content and style separately with self-supervised guidance. By disentangling the discriminator for each information, it can be possible to train the converted speech which does not have ground-truth audio. It is worth noting again that using other supervised features (e.g., pitch contour or text transcription) help the model to disentangle each information, but our model uses self-supervised representation learning without additional features. We describe the details of our architecture, similarity-based information bottleneck, and the adversarial voice style mixup in the following subsections.

### 3.1 Generator

For the generator, we follow the autoencoder framework of AUTOVC. As shown in Figure 1a, the generator $G$ consists of a content encoder $f_c(\cdot)$ which extracts the content embedding from speech, a speaker encoder $f_s(\cdot)$ which extracts a speaker embedding from speech, and a decoder $g(\cdot, \cdot)$ which generates the speech from content and speaker embeddings represented in Equation 1.

### 3.2 Similarity-based information bottleneck

Unlike information bottleneck in Equation 2, we downsample the content embedding according to the similarity between the content embeddings. We assume that the content encoder produces similar content embedding from similar phoneme, and thus we downsample the adjacent phonetic information to be mapped together. We calculate the similarity $Q = (q_1, \cdots, q_T)$ between content embedding sequence $C = (c_1, \cdots, c_T)$ and shifted content embedding sequence $C_{shift} = (c_2, \cdots, c_{T+1})$ as:

$$q_t = sig(\frac{c_t \cdot c_{t+1}}{\|c_t\| \|c_{t+1}\|}/\rho), \tag{3}$$

where $sig$ denotes the sigmoid function and $\rho$ is the temperature parameter. Then, we extract the similarity-based duration $\mathcal{D} = (d_1, \cdots, d_N)$ where $d_n$ is cumulative sum until the similarity $q_t$ is under the average similarity, and the $d_{n+1}$ is computed again from $q_{t+1}$ until time step $T$.

**Gaussian down/up-sampling**  Assume that the center of same content have the largest information of that content, then we apply the Gaussian downsampling to focus attention to the center. Given the content embedding to be downsampled $C$, duration $\mathcal{D}$, and learnable range parameter $\sigma = (\sigma_1, \cdots, \sigma_N)$ like (Shen et al., 2020), we compute downsampled sequence $H = (h_1, \cdots, h_N)$ as:

$$\alpha_n = \frac{d_n}{2} + \sum_{m=1}^{n-1} d_m, \quad w_t^n = \frac{\mathcal{N}(t; \alpha_n, \sigma_n^2)}{\sum_{m=1}^{N} \mathcal{N}(t; \alpha_m, \sigma_m^2)}, \quad h_n = \sum_{t=1}^{T} w_t^n c_t \tag{4}$$

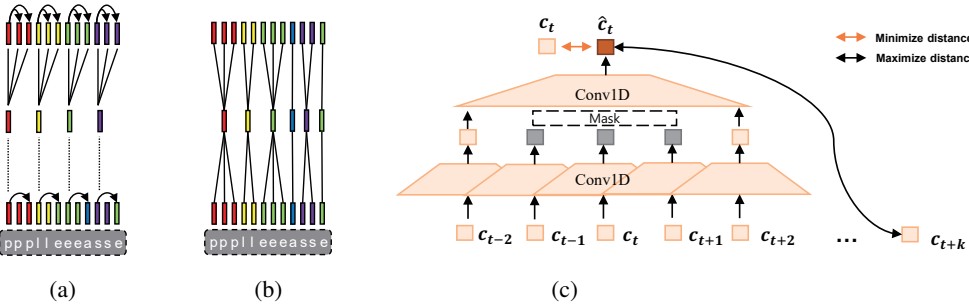

Figure 2: (a) Fixed-length information bottleneck. (b) Similarity-based information bottleneck. (c) Context network for self-supervised representation learning on content embedding.

Afterwards, we use Gaussian upsampling as a TTS model following (Shen et al., 2020) to upscale $\boldsymbol{H}$ to upsampled content sequence $\tilde{\boldsymbol{C}} = (\tilde{\boldsymbol{c}}_1, \cdots, \tilde{\boldsymbol{c}}_T)$ with the same duration of $\mathcal{D}$, range parameter for upsampling $\boldsymbol{\sigma}' = (\sigma_1', \cdots, \sigma_N')$, and then $\tilde{\boldsymbol{C}}$ is fed to $g(\cdot, \cdot)$ to generate the mel-spectrogram as:

$$w_t'^n = \frac{\mathcal{N}(t; \alpha_n, \sigma_n'^2)}{\sum_{m=1}^{N} \mathcal{N}(t; \alpha_m, \sigma_m'^2)}, \quad \tilde{\boldsymbol{c}}_t = \sum_{n=1}^{N} w_t'^n \boldsymbol{h}_n, \quad \hat{\boldsymbol{X}}_{1 \to 1, A} = g(\boldsymbol{S_1}, \tilde{\boldsymbol{C}}_A) \tag{5}$$

### 3.3 Auxiliary losses for similarity

**Contrastive loss** To increase the similarity between the adjacent content embeddings, we train the content encoder with self-supervised representation learning. The content embedding is fed to a context network $f_r$ to learn a content representation illustrated in Figure 2c. To train in non-autoregressive manner, we utilize the masked convolutional blocks (Liu et al., 2020) to predict a content embedding from the adjacent content embeddings, and the contrastive loss for positive sample is defined to minimize distance between predicted content embedding $\hat{\boldsymbol{C}} = (\hat{\boldsymbol{c}}_1, \cdots, \hat{\boldsymbol{c}}_T)$ and content embedding $\boldsymbol{C}$:

$$\mathcal{L}_{pos}(f_c, f_r) = \mathbb{E}\Big[ - \frac{1}{T} \sum_{i}^{T} \log sig(\frac{\boldsymbol{c}_i \cdot \hat{\boldsymbol{c}}_i}{\|\boldsymbol{c}_i\| \|\hat{\boldsymbol{c}}_i\|}/\rho) \Big] \tag{6}$$

where $sig$ denotes the sigmoid function and $\rho$ represents the temperature parameter.

To remove style information on the content embedding in an unsupervised manner, we prevent context network to predict future representation of content embedding. While negative samples are uniformly sampled from the same utterance in (Baevski et al., 2020), we only sample the $k$-th future content embedding as a negative sample to prevent maximizing distance between the contents similar to each other. We maximize cosine distance between predicted content embedding and $k$-th future representation of content embedding, and the contrastive loss for negative sample is:

$$\mathcal{L}_{neg}(f_c, f_r) = \mathbb{E}\Big[ \frac{1}{T} \sum_{i}^{T} \log sig(\frac{\boldsymbol{c}_{i+k} \cdot \hat{\boldsymbol{c}}_i}{\|\boldsymbol{c}_{i+k}\| \|\hat{\boldsymbol{c}}_i\|}/\rho) \Big]. \tag{7}$$

**Adversarial speaker classification** To enforce speaker disentanglement on the content embedding, we apply adversarial speaker classification in a supervised manner (using speaker label $\boldsymbol{y}_i$) as:

$$\mathcal{L}_{advsc}(f_c) = \mathbb{E}\Big[ \frac{1}{T} \sum_{i}^{T} \boldsymbol{y}_i log(f_{cls}(\boldsymbol{c}_i)) \Big] \tag{8}$$

where $f_{cls}$ denotes speaker classifier. To train the entired model jointly with $f_{cls}$, we use a gradient reversal layer before the content embedding is fed to $f_{cls}$ following (Hsu et al., 2019).

### 3.4 Disentangled discriminator with self-supervised guidance

Unlike the previous GAN-based VC model which uses the cycle-consistency training to preserve linguistic information by two-way generation, we follow the autoencoder based reconstruction method

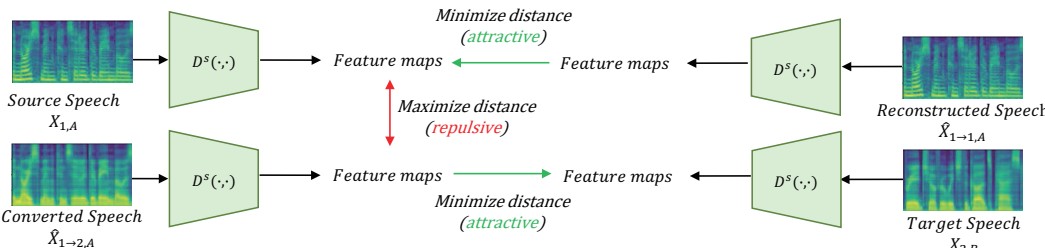

Figure 3: Style feature matching loss for reconstructed and converted mel-spectrogram.

for training. For adversarial feedback, we divide the discriminator $D$ into the content discriminator $D^c(\cdot, \cdot)$ and style discriminator $D^s(\cdot, \cdot)$ to disentangle content and style, respectively. To guide each discriminator for each attribute, we condition the content embedding to the content discriminator and style embedding to the style discriminator as a self-supervised conditional information illustrated in Figure 1. For the training objectives, we use the LSGAN (Mao et al., 2017) as followed:

$$\mathcal{L}_{adv}^D(D^c, D^s; G) = \mathbb{E}\Big[\|D^c(\boldsymbol{X}_{1,A}, \boldsymbol{C}_A) - 1\|_2 + \|D^c(\hat{\boldsymbol{X}}_{1\to1,A}, \boldsymbol{C}_A)\|_2$$
$$+ \|D^s(\boldsymbol{X}_{1,A}, S_1) - 1\|_2 + \|D^s(\hat{\boldsymbol{X}}_{1\to1,A}, S_1)\|_2\Big] \tag{9}$$

$$\mathcal{L}_{adv}^G(G; D^c, D^s) = \mathbb{E}\Big[\|D^c(\hat{\boldsymbol{X}}_{1\to1,A}, \boldsymbol{C}_A) - 1\|_2 + \|D^s(\hat{\boldsymbol{X}}_{1\to1,A}, \boldsymbol{S}_1) - 1\|_2\Big] \tag{10}$$

**Feature Matching for reconstruction** We use the feature matching loss to train the generator, which minimizes the distance of the discriminator's features between ground truth and generated speech. For each discriminator, we use the content feature matching loss $\mathcal{L}_{content}^G$ from the content discriminator for content and the style feature matching loss $\mathcal{L}_{style}^G$ from the style discriminator for style.

$$\mathcal{L}_{content}^G(G; D^c) = \mathbb{E}\bigg[\sum_{i=1}^K \frac{1}{N_i}\|D_i^c(\boldsymbol{X}_{1,A}, \boldsymbol{C}_A) - D_i^c(\hat{\boldsymbol{X}}_{1\to1,A}, \boldsymbol{C}_A)\|_1\bigg] \tag{11}$$

$$\mathcal{L}_{style}^G(G; D^s) = \mathbb{E}\bigg[\sum_{i=1}^{K'} \frac{1}{N_i}\|D_i^s(\boldsymbol{X}_{1,A}, \boldsymbol{S}_1) - D_i^s(\hat{\boldsymbol{X}}_{1\to1,A}, \boldsymbol{S}_1)\|_1\bigg] \tag{12}$$

where $K$ and $K'$ denote the number of blocks in each discriminator, and $N_i$ is the number of features in $i$-th discriminator block. The total loss for reconstructed mel-spectrogram is defined as:

$$\mathcal{L}_{rec} = \mathcal{L}_{adv}^G(G; D^c, D^s) + \lambda_c\mathcal{L}_{content}^G(G; D^c) + \lambda_s\mathcal{L}_{style}^G(G; D^s) + \lambda_{mel}\mathcal{L}_{mel} \tag{13}$$

where $\mathcal{L}_{mel}$ is mean absolute error between $X_{1,A}$ and $\hat{X}_{1\to1,A}$.

### 3.5 Adversarial Voice Style Mixup

By introducing the disentangled discriminator for each information, we can train the reconstructed speech for each disentangled feature. However, our goal is to convert voice by disentangling the source style and transferring the target style. To learn the latent representations of the converted speech, we propose an adversarial voice style mixup, which can train the converted speech by using the disentangled discriminator with a self-supervised condition. Even though converted speech does not have ground-truth (GT) samples, the converted mel-spectrogram can be trained with adversarial feedback through each discriminator. It is worth noting that the model only uses a self-supervised hidden representation of the generator as conditional features without any external feature extractor for conditional information. The GAN losses for the converted mel-spectrogram are defined as:

$$\mathcal{L}_{adv^*}^D(D^c, D^s; G) = \mathbb{E}\Big[\|D^c(\hat{\boldsymbol{X}}_{1\to2,A}, \boldsymbol{C}_A)\|_2 + \|D^s(\hat{\boldsymbol{X}}_{1\to2,A}, \boldsymbol{S}_2)\|_2\Big], \tag{14}$$

$$\mathcal{L}_{adv^*}^G(G; D^c, D^s) = \mathbb{E}\Big[\|D^c(\hat{\boldsymbol{X}}_{1\to2,A}, \boldsymbol{C}_A) - 1\|_2 + \|D^s(\hat{\boldsymbol{X}}_{1\to2,A}, \boldsymbol{S}_2) - 1\|_2\Big] \tag{15}$$

**Feature Matching for Mixup**    We can also use the feature matching loss for a converted speech by the disentangled discriminator. For the content representation, the feature distance of content discriminator between converted speech and source speech is minimized as:

$$\mathcal{L}^G_{content*}(G; D^c) = \mathbb{E}\bigg[\sum_{i=1}^{K} \frac{1}{N_i} \|D^c_i(\boldsymbol{X}_{1,A}, \boldsymbol{C}_A) - D^c_i(\hat{\boldsymbol{X}}_{1\to2,A}, \boldsymbol{C}_A)\|_1\bigg] \qquad (16)$$

For the style representation, the feature distance of style between converted and target speech is minimized as following:

$$\mathcal{L}^G_{style+}(G; D^s) = \mathbb{E}\bigg[\sum_{i=1}^{K'} \frac{1}{N_i} \|D^s_i(\boldsymbol{X}_{2,B}, \boldsymbol{S}_2) - D^s_i(\hat{\boldsymbol{X}}_{1\to2,A}, \boldsymbol{S}_2)\|_1\bigg] \qquad (17)$$

We call it *"Attractive style loss"* which minimizes the style feature distance between the same style of the same speaker. As shown in Figure 3, we also introduce *"Repulsive style loss"* to maximize the style feature distance between the different style of the converted speech and source speech as:

$$\mathcal{L}^G_{style-}(G; D^s) = \mathbb{E}\bigg[-\sum_{i=1}^{K'} \frac{1}{N_i} \|D^s_i(\boldsymbol{X}_{1,A}, \boldsymbol{S}_1) - D^s_i(\hat{\boldsymbol{X}}_{1\to2,A}, \boldsymbol{S}_2)\|_1\bigg] \qquad (18)$$

When the content encoder does not disentangle the source speaker, the converted speech may contain the style of the source speaker. Thus, this repulsive style loss restricts the converted speech from having the style of source speaker. The total loss for converted mel-spectrogram is defined as:

$$\mathcal{L}_{con} = \mathcal{L}^G_{adv*}(G; D^c, D^s) + \lambda_c\mathcal{L}^G_{content*}(G; D^c) + \lambda_s\mathcal{L}^G_{style+}(G; D^s) + \lambda_{s^-}\mathcal{L}^G_{style-}(G; D^s) \quad (19)$$

Our final objectives for the discriminators and generator are represented as:

$$\mathcal{L}^D = \mathcal{L}^D_{adv}(D^c, D^s; G) + \lambda_{con}\mathcal{L}^D_{adv*}(D^c, D^s; G) \qquad (20)$$

$$\mathcal{L}^G = \mathcal{L}_{rec} + \lambda_{con}\mathcal{L}_{con} + \lambda_{pos}\mathcal{L}_{pos}(f_c, f_r) + \lambda_{neg}\mathcal{L}_{neg}(f_c, f_r) + \lambda_{advsc}\mathcal{L}_{advsc}(f_c) \qquad (21)$$

## 4    Experiment and result

We evaluated our model with the VCTK dataset, which has 46 hours of audio from 109 speakers (Veaux et al., 2017). We divided the dataset into 98 speakers as base speakers for many-to-many VST and 10 speakers as the novel speakers for zero-shot VST. The base speaker is split into train and test sets. For the non-parallel dataset setting, the training set consists of different utterances for all of the speakers, and the test set consists of 25 same utterances. We transform the mel-spectrogram with 80 bins from the audio downsampled at 22,050Hz. The spectrogram is inverted to a waveform by the pre-trained HiFi-GAN (Kong et al., 2020). For many-to-many VST, we randomly choose 20 speakers with equal distribution of male and female from the base speakers. For zero-shot VST, we randomly choose 10 speakers from the base speakers, and 10 speakers from the novel speakers. For each setting, a single utterance is selected from each speaker, and then all the possible pairs of utterances (20 x 20 = 400) are produced.

### 4.1    Implementation details

The generator consists of a speaker encoder, content encoder, similarity-based information bottleneck, and decoder. We train the entire model jointly. The speaker embedding is extracted from the speaker encoder which has the same architecture as the reference encoder in (Skerry-Ryan et al., 2018). The source speech is fed to the content encoder consisting of a pre-net and three blocks of the multi-receptive field fusion (MRF) (Kong et al., 2020). The pre-net is two linear layers with 384 channels. The output concatenated with source speaker embedding is fed to a 1D convolutional layer with 384 channels, followed by the MRF. The MRF returns the sum of output from 384 channels of multiple convolutional layers with multiple dilations and multiple receptive fields. We use the combination of two dilations of [1, 3], and two receptive fields of [3, 7] for the MRF. Before the features are fed to MRF, the instance normalization (IN) is applied.

The similarity-based information bottleneck has two range predictors for down/up-sampling. Both range predictors consist of three convolutional layers followed by a linear layer with a softplus activation function, similar to (Shen et al., 2020). The range predictor for downsampling uses the similarity-based duration as input. The range predictor for upsampling uses the same duration and downsampled content embedding as input. The content embedding is fed to both adversarial speaker classifier and contrastive encoder. The adversarial speaker classifier consists of five 1D convolutional layers followed by a linear layer to predict speaker identity. To remove the speaker identity on content embedding, a gradient reversal layer is used before the first layer of the adversarial speaker classifier. For the contrastive encoder, we use three masked convolution blocks of (Liu et al., 2020) with 384 channels, receptive field size of 23, and mask sizes of [5, 7, 9]. We set the $k$ as 24 (about 0.3s) which is over the average duration of consonant-vowel syllables (Steinschneider et al., 2013).

After the similarity-based information bottleneck, the upsampled feature concatenated with the target speaker embedding is fed to the decoder, which consists of a conditional layer, three blocks of MRF, and the linear layer. The conditional layer is a single 1D convolutional layer with 384 channels. The MRF of the decoder has the same architecture with the MRF of the encoder without IN applied to the feature before being fed to the MRF. Finally, the mel-spectrogram is predicted by the linear layer.

The content discriminator consists of four blocks which have a speech-side and content-condition side block following (Lee et al., 2021). Each block has two 1D convolutional layers. The hidden representation of the condition-side block is added to the speech-side hidden representations of [256, 512, 1024, 1024]. The style discriminator consists of 4 blocks which have a speech-side block and style-condition side linear layers. Each output of the linear layer is added to the speech-side hidden representations of [256, 512, 1024, 1024]. We report more details of hyperparameter in Appendix A.

## 4.2 Information bottleneck alignment

To compare the similarity-based information bottleneck using a different combination of auxiliary loss, Figure 4 show the alignment between source speech and downsampled content embedding. The model without contrastive loss ($\mathcal{L}_{pos}$ and $\mathcal{L}_{neg}$) nearly shows the diagonal alignment, which implies that content embedding does not only represent content information. The model without $\mathcal{L}_{advsc}$ shows alignment closer to attention alignment of Tacotron2 (Shen et al., 2018). These results show that contrastive loss is more important to disentangle content and speaker information than $\mathcal{L}_{advsc}$. Without using any text transcript and target duration, our model with auxiliary losses produces an alignment similar to phonetic alignment, and shows better performance as shown in Table 3.

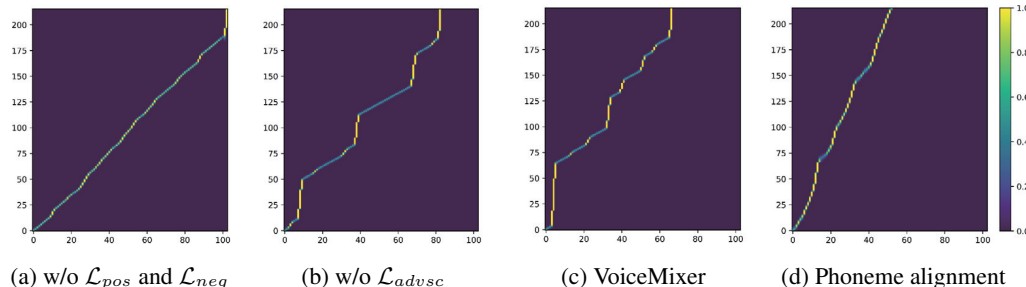

| (a) w/o $\mathcal{L}_{pos}$ and $\mathcal{L}_{neg}$ | (b) w/o $\mathcal{L}_{advsc}$ | (c) VoiceMixer | (d) Phoneme alignment |

Figure 4: Alignment of similarity-based information bottleneck

## 4.3 Evaluation metrics

**Subjective metrics**  We conduct the naturalness and similarity mean opinion score test. For the naturalness of speech (Naturalness), converted samples are evaluated by at least 20 raters on a scale of 1 to 5. The Naturalness is reported with 95% confidence intervals. For the similarity of converted speech (Similarity), both converted and target speech are presented to at least 20 raters, and the raters evaluate on a scale of 1 to 4. We also report the binary rating introduced in (Serrà et al., 2019).

**Objective metrics**  We conduct three objective metrics; the equal error rate of the automatic speaker verification (ASV EER), the mel-cepstral distance (MCD$_{13}$) (Kubichek, 1993), and the $F0$ root mean square error (RMSE$_{f0}$). We use the pre-trained ASV model (Chung et al., 2020) trained by the large

Table 1: Many-to-many VST evaluation results.

| Method | Naturalness | Similarity | ASV EER | $MCD_{13}$ | $RMSE_{f0}$ | FDSD |
|---|---|---|---|---|---|---|
| GT | 4.07±0.03 | 93.7% | 5.5% | - | - | - |
| HiFi-GAN (Vocoder) | 3.92±0.03 | 92.8% | 7.6% | 3.19 | 36.22 | 0.546 |
| StarGAN-VC | 3.48±0.04 | 42.5% | 22.7% | 7.97 | 33.98 | 13.561 |
| AGAIN-VC | 3.60±0.03 | 52.2% | 14.3% | 6.75 | 41.39 | 3.185 |
| AUTOVC ($\tau$=16) | 3.65±0.03 | 52.1% | 18.3% | 6.70 | 44.05 | 5.753 |
| AUTOVC ($\tau$=32) | 3.64±0.03 | 52.1% | 14.0% | 6.44 | 39.93 | 10.703 |
| AUTOVC + $\mathcal{L}_{advsc}$ ($\tau$=16) | 3.63±0.03 | 54.6% | 14.7% | 6.58 | 39.40 | 6.036 |
| Blow | 3.12±0.04 | 33.2% | 52.0% | 6.74 | 44.55 | 15.112 |
| VoiceMixer (Ours) | 3.78±0.03 | 55.9% | 12.5% | 6.77 | 42.76 | 2.080 |

Table 2: Zero-shot VST evaluation results.

| Method | Naturalness | Similarity | ASV EER | $MCD_{13}$ | $RMSE_{f0}$ | FDSD |
|---|---|---|---|---|---|---|
| GT | 4.08±0.03 | 96.3% | 4.4% | - | - | - |
| HiFi-GAN (Vocoder) | 4.03±0.03 | 95.4% | 6.0% | 3.34 | 38.33 | 0.576 |
| AGAIN-VC | 3.29±0.03 | 58.2% | 15.0% | 6.96 | 44.81 | 3.261 |
| AUTOVC ($\tau$=16) | 3.40±0.03 | 46.3% | 25.0% | 6.92 | 46.33 | 5.227 |
| AUTOVC ($\tau$=32) | 3.24±0.03 | 59.7% | 20.7% | 6.65 | 39.93 | 9.741 |
| AUTOVC + $\mathcal{L}_{advsc}$ ($\tau$=16) | 3.39±0.03 | 58.9% | 21.9% | 6.78 | 42.08 | 5.675 |
| VoiceMixer (Ours) | 3.75±0.03 | 63.4% | 18.5% | 7.04 | 44.70 | 2.416 |

scale dataset, VoxCeleb2 (Chung et al., 2018). We compute the EER from the converted and target speech (400 x 20 = 8000). We apply the dynamic time warping (DTW) to calculate the $MCD_{13}$ and $RMSE_{f0}$ between converted and target speech, which has different time alignment. For the naturalness measurement, we evaluate Fréchet DeepSpeech Distance (FDSD) (Bińkowski et al., 2020; Gritsenko et al., 2020), which is the distance between the high-level features of GT and generated audio from the pre-trained DeepSpeech2 (Amodei et al., 2016). Because DeepSpeech2 is a speech recognition model trained with connectionist temporal classification loss to classify the text sequence, the hidden representations are related to linguistic information. Thus, we use FDSD between the converted and source speech for objective naturalness measurement.

## 4.4 Audio quality and style transfer performance

For the many-to-many VST evaluation, we compared our model with four VC models; StarGAN-VC (Kameoka et al., 2018), AGAIN-VC (Chen et al., 2020), Blow (Serrà et al., 2019), and AUTOVC (Qian et al., 2019). All models are trained on the same dataset as VoiceMixer, and the implementation details are described in the Appendix A. We trained AUTOVC in various settings with each having different sizes of information bottleneck. For a fair comparison, we also implemented the AUTOVC model with an adversarial speaker classifier. Table 1 shows that our model outperforms other models in Naturalness and FDSD metrics. Our model also shows better transfer performance on the Similarity and ASV EER. When disentangling the content and style information, AGAIN-VC and AUTOVC models lose a lot of content information, and thus the converted speech has lower Naturalness and higher FDSD score. The adversarial speaker classifier on the content embedding can help the disentanglement performance of models, but the naturalness can be degraded.

For the zero-shot VST evaluation, we compared our model with two VC models; AGAIN-VC and AUTOVC as shown in Table 2. We also implemented various AUTOVC trained with different information bottleneck size and adversarial speaker classification loss. Our model has better performance in Naturalness and FDSD. In terms of similarity, even though the AGAIN-VC has higher performance in ASV EER, our model has better performance in Similarity. In terms of AUTOVC, it is hard to select the proper down-sampling factor, which has a trade-off between naturalness and similarity. Thus, it is important to note that our proposed similarity-based information bottleneck need not find the proper factor, which is learned by self-supervised representation learning with a small loss of content information. Our audio samples are available on the demo page.[2]

---

[2] `https://anonymous-speech.github.io/voicemixer/index.html`

Table 3: Ablation studies for zero-shot voice style transfer

| Method | Naturalness | Similarity | ASV EER | $MCD_{13}$ | $RMSE_{f0}$ | FDSD |
|---|---|---|---|---|---|---|
| VoiceMixer (Ours) | $3.72\pm0.03$ | 63.7% | 18.5% | 7.04 | 44.70 | 2.416 |
| w/o $\mathcal{L}_{advsc}$ | $3.74\pm0.03$ | 41.3% | 32.2% | 7.33 | 47.63 | 1.217 |
| w/o $\mathcal{L}_{pos}$ and $\mathcal{L}_{neg}$ | $3.21\pm0.03$ | 42.3% | 38.0% | 7.92 | 54.11 | 4.634 |
| w/o $\mathcal{L}_{neg}$ | $3.18\pm0.03$ | 45.5% | 24.2% | 7.72 | 45.05 | 11.354 |
| w fixed-length IB ($\tau$=16) | $3.35\pm0.03$ | 51.8% | 28.5% | 7.38 | 46.14 | 2.827 |
| w fixed-length IB ($\tau$=32) | $3.28\pm0.03$ | 56.8% | 20.7% | 6.97 | 44.16 | 2.823 |
| w/o GAN | $3.70\pm0.03$ | 58.5% | 20.5% | 6.95 | 44.79 | 2.666 |
| w/o disentangled discriminator | $3.68\pm0.03$ | 60.4% | 18.3% | 7.02 | 45.23 | 3.502 |
| w/o $\mathcal{L}^*_{style^-}$ | $3.69\pm0.03$ | 55.0% | 21.6% | 7.02 | 45.66 | 2.173 |

## 4.5 Ablation study

We conducted ablation studies for the information bottleneck and adversarial feedback in Table 3. We evaluate each model for the same zero-shot VST setting of Table 2. In terms of similarity-based information bottleneck, absence of $\mathcal{L}_{advsc}$ or using fixed-length information bottleneck (IB) makes it harder to disentangle the content and style in content embedding. For better disentanglement, it is essential to use $\mathcal{L}_{pos}$ with $\mathcal{L}_{neg}$, and it induces the information bottleneck to downsample the content embedding similar to the phoneme alignment as shown in Figure 4. Additionally, when trained without any information bottleneck or all of the auxiliary losses, these models are not able to convert any voice, but only reconstruct source speech. Using GAN makes the model have better performance on all of the metrics. The model trained with a single discriminator (instead of disentangled discriminator) shows lower performance in FDSD. In this regard, disentangled discriminator encourages better generalization for each attribute. Removing $\mathcal{L}^*_{style^-}$ results in lower performance in both subjective and objective similarity metrics even though the FDSD decreases.

## 4.6 Content and speaker disentanglement

We conduct speaker classification on content embedding to evaluate the disentanglement performance compared to AGAIN-VC and AUTOVC. Table 4 represents the classification results. Our model shows the lowest accuracy, which mean our model has better disentanglement performance by removing speaker identity on content embedding despite having the largest feature dimensions of 384 (The content embedding of AUTOVC has 64 dimensions). Even though AUTOVC trained with adversarial speaker classifier has comparable disentanglement performance, AUTOVC loses a lot of content information in their fixed-length based information bottleneck as shown in Table 1. AUTOVC also has to heuristically find optimal

Table 4: Speaker classification accuracy on content embedding of the autoencoder based VC models and word error rate (WER) of converted speech.

| Method | ACC | WER |
|---|---|---|
| GT | - | 8.18 |
| HiFi-GAN (Vocoder) | - | 10.12 |
| AGAIN-VC | 27.31 | 27.26 |
| AUTOVC ($\tau$=16) | 10.27 | 39.27 |
| AUTOVC ($\tau$=32) | 4.47 | 61.99 |
| AUTOVC + $L_{advsc}$ ($\tau$=16) | 3.11 | 39.31 |
| VoiceMixer (Ours) | 1.47 | 20.92 |

downsampling size for a good balance between content and style. On the other hand, our proposed model finds proper downsampling size based on the similarity of content embedding learned by self-supervised representation learning with only a small loss of content information. We additionally conducted word error rate (WER) evaluation on the converted speech. The results show VoiceMixer has a lower WER of 20.92%, which support our model converts the voice with a small loss of content information. We used the Google Speech-to-Text API for ASR model.

## 5 Broader Impact

We see voice conversion technology could be used for various applications in many areas. First, the personalized voice systems for the entertainment industry can be utilized in game or voice chat.

This may help to secure the privacy of the player. Furthermore, it is also possible to read books with parents' voices for their children by converting the audiobook. The film industry can also use the system for dubbing the original actors' voice in different languages. Finally, voice conversion can also restore the voice for individuals who passed away or have lost their ability to speak by a neurological disorder or motor disabilities such as amyotrophic lateral sclerosis or spinal cord injury. However, we also acknowledge the potential harms, malicious use, and ethical concerns of voice conversion technology against the positive impacts.

**Social negative impact and mitigation strategy**   As speech-related AI systems such as voice conversion and text-to-speech synthesize realistic audio, there is an increased risk of the potential harms, malicious use, and ethical concerns for TTS and VC systems. These systems can be used to deceive people in various ways; usage in crimes such as voice spoofing (Kinnunen et al., 2012), fake news (Singhal et al., 2019), and commercial use by cloning voice without consent (Wang et al., 2020b). Also, there is a potential use for fraud, theft, abuse, or harassment. Hence, it is necessary to consider a countermeasure for reducing these potential risks. The anti-spoofing techniques (Wu et al., 2012) such a fake audio detection (Tak et al., 2021; Mittal et al., 2020; Ma et al., 2021) are developed to distinguish synthesized speech, and (Müller et al., 2021) demonstrated that the AI outperforms the human in fake audio attacks. Moreover, recently, there is an attack that uses a partially fake audio clip, and some small fake audio clips are hidden in real speech audio. As (Yi et al., 2021) indicated that partially fake audio detection is more challenging than fully fake audio, it is also important to develop a partially fake audio detection model. For practical mitigation strategies, there are some potential approaches. First, voice conversion system should be deployed when conditioned on written consent from all parties involved (e.g., the speakers of source and target speech) if the dataset is not open-sourced or licensed to be open publicly. This can prevent inappropriate use of voice conversion without agreement from affiliated parties. If either of the two parties is dead or unavailable, there must be other measures taken such as consent from the direct family. Second, the converted speech or synthesized speech should be provided when conditioned on mandatory disclosure with the statement such as *"this speech is synthesized by voice conversion or text-to-speech model"*. This can potentially prevent harmful and malicious conduct such as voice spoofing, phishing, or faking using voice conversion. Third, the user could synthesize the speech with the virtual voice by mixing the style vector from the multiple speakers where that voice doesn't exist in real-world. However, the user should use the data for style information, given the written consent from all of the parties. Finally, there is a risk that the system trained with the leaked private data may be used to deceive people, so researcher and institution should enhance their privacy protection mechanism. To do this, (Hong et al., 2021) introduced federated learning for text-to-speech. Although these mitigation strategies are still imperfect, we will continue to make improvements on mitigation strategies and dedicate thoughtful discussions on methods to prevent harmful use in our future works.

# 6   Conclusion

We presented VoiceMixer, which can decompose and transfer voice style by similarity-based information bottleneck and adversarial feedback with self-supervised representation learning. Without effort to find the proper size of information bottleneck carefully, our model is able to learn proper downsampling factor with self-supervised representation learning. We successfully demonstrated that our novel information bottleneck can decompose content and style information from the source speech with a small loss of content information. Moreover, the alignment of information bottleneck is similar to phonetic alignment despite not using any text transcript and target phoneme duration. we also show the adversarial voice style mixup makes it possible to learn the latent representation of converted speech which does not have ground-truth speech, and it improves the overall generalization.

While there remains a gap between target and converted voice, we believe pre-trained speaker encoder with large-scale dataset (Jia et al., 2018) could improve the VST performance. Moreover, we see our self-supervised learning based speech disentanglement extending to other tasks. For future work, we will apply our speech disentanglement to TTS without text (Dunbar et al., 2019; Tan et al., 2021), which is a challenging task to synthesize the speech without any text transcript for use of the untranscribed large-scale speech data. We will also attempt to control the rhythm of speech like (Qian et al., 2020, 2021). We discussed the potential positive and negative impact of voice conversion in section 5. We hope that our method will be used in positive applications or extend to technical mitigation strategy such as fake audio detection.

## Acknowledgements

This work was partly supported by Institute of Information & communications Technology Planning & Evaluation (IITP) grant funded by the Korea government (MSIT) (No. 2019-0-00079, Artificial Intelligence Graduate School Program (Korea University) and No. 2021-0-02068, Artificial Intelligence Innovation Hub) and Microsoft Research Asia (MSRA).

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
