# A Implementation details

For the reproducibility, we provided the source code of model. For the request of Neurips2021 committee, however, it is restricted by private repository via request[3]. We train the model using the AdamW optimizer (Loshchilov and Hutter, 2019) with $\beta_1 = 0.8$, $\beta_2 = 0.99$, and weight decay $\lambda = 0.01$, and apply the learning rate schedule as that of (Kong et al., 2020) with initial learning rate of $2 \times 10^{-4}$ for generator and $2 \times 10^{-6}$ for discriminator. We train VoiceMixer with a batch size of 64 for 150k steps. The architecture of VoiceMixer is illustrated in Figure 5, 6.

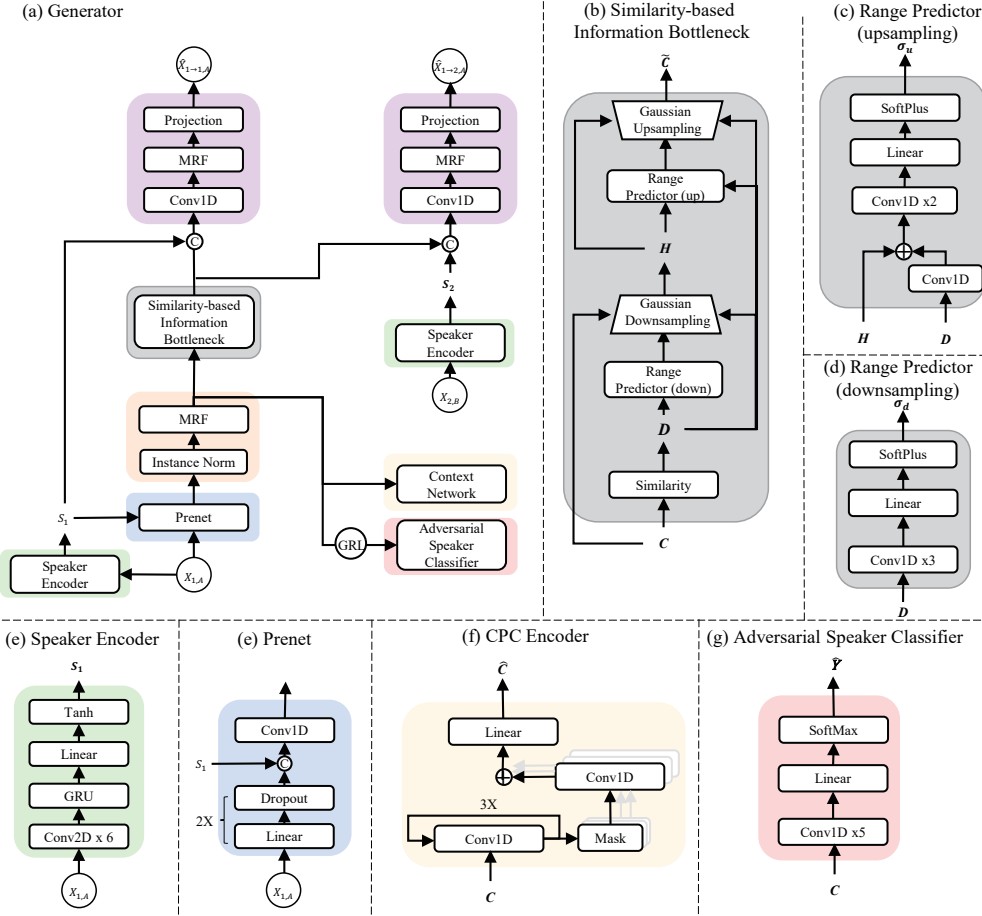

Figure 5: Generator architecture of VoiceMixer

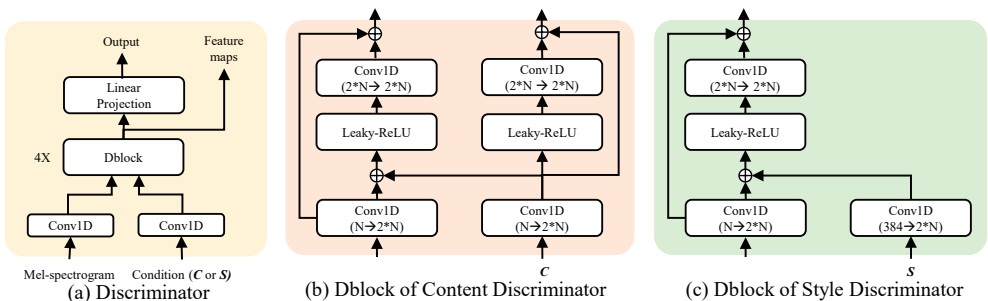

Figure 6: Discriminator architecture of VoiceMixer

[3]https://github.com/anonymous-speech/voicemixer/tree/main/code

Table 5: Hyperparmeters of generator

| Hyperparameter | VoiceMixer |
|---|---|
| Mel-spectrogram Dimension | 80                                           - |
| Prenet Linear Layer | 2 |
| Prenet Linear Hidden | 384 |
| Prenet Conv1D Layer | 1 |
| Prenet Conv1D Hidden | 384 |
| Prenet Conv1D Kernel | 7 |
| Speaker Encoder Conv1D Layer | 6 |
| Speaker Encoder Conv1D Hidden | [32,64,128,192,256,384] |
| Speaker Encoder Conv1D Kernel | 3 |
| Speaker Encoder Conv1D Stride | 2 |
| Speaker Encoder GRU Hidden | 384 |
| Speaker Encoder Linear projection | 384 |
| Content Encoder MRF Module | 3 |
| Content Encoder MRF kernel | [3, 7] |
| Content Encoder MRF dilation | [[1, 3], [1,3]] |
| Content Encoder MRF Filter Size | 384 |
| Decoder Conv1D Layer | 1 |
| Decoder Conv1D Hidden | 384 |
| Decoder MRF Module | 3 |
| Decoder MRF Kernel | [3, 7] |
| Decoder MRF Dilation | [[1, 3], [1,3]] |
| Decoder MRF Filter Size | 384 |
| Decoder Linear Projection | 80 |
| Range Predictor Conv1D layer | 2 |
| Range Predictor Conv1D Kernel | 3 |
| Range Predictor Conv1D Filter size | 384 |
| Range Predictor Dropout Size | 0.5 |
| Context Network Conv1D layer | 3 |
| Context Network Conv1D kernel | 3 |
| Context Network Conv1D Filter | 384 |
| Context Network Conv1D Dropout | 0.5 |
| Context Network MaskedConv1D layer | 3 |
| Context Network MaskedConv1D kernel | 23 |
| Context Network MaskedConv1D Mask | [5,7,9] |
| Context Network MaskedConv1D Filter | 384 |
| Context Network Linear Projection | 384 |
| Adversarial Speaker Classifier Conv1D layer | 5 |
| Adversarial Speaker Classifier Conv1D Kernel | 3 |
| Adversarial Speaker Classifier Conv1D Filter Size | 256 |
| Adversarial Speaker Classifier Conv1D stride | 2 |
| Adversarial Speaker Classifier Dropout Size | 0.1 |
| Encoder/Decoder Dropout | 0.2 |
| k | 24 |
| $\rho$ | 0.1 |
| Batch Size | 64 |
| $\lambda_{adv}/\lambda_c/\lambda_s/\lambda_{s-}$ | 0.01/0.02/0.02/0.04 |
| $\lambda_{mel}/ \lambda_{con}/ \lambda_{pos}/ \lambda_{neg}/\lambda_{advsc}$ | 45/1/45/9/1 |

Table 6: Hyperparameter of discriminator

| Hyperparameter | VoiceMixer |
|---|---|
| Content Discriminator Conv1D layer | 1 |
| Content Discriminator Conv1D kernel | 3 |
| Content Discriminator Conv1D filter | 256 |
| Content Discriminator Blocks First Conv1D Input | [256, 256, 512, 1024] |
| Content Discriminator Blocks First Conv1D Filter | [256, 512, 1024, 1024] |
| Content Discriminator Blocks First Conv1D Kernel | 3 |
| Content Discriminator Blocks First Conv1D Stride | [1,2,2,2] |
| Content Discriminator Blocks Second Conv1D Filter | [256, 512, 1024, 1024] |
| Content Discriminator Blocks Second Conv1D Kernel | 1 |
| Content Discriminator Blocks First Conv1D Stride | 1 |
| Style Discriminator Conv1D layer | 1 |
| Style Discriminator Conv1D kernel | 3 |
| Style Discriminator Conv1D filter | 128 |
| Style Discriminator Spectrogram-side Block First Conv1D Input | [128, 256, 512, 1024] |
| Style Discriminator Spectrogram-side Block First Conv1D Filter | [256, 512, 1024, 1024] |
| Style Discriminator Spectrogram-side Block First Conv1D Kernel | 9 |
| Style Discriminator Spectrogram-side Block First Conv1D Stride | [2,2,2,2] |
| Style Discriminator Spectrogram-side Block Second Conv1D Filter | [256, 512, 1024, 1024] |
| Style Discriminator Spectrogram-side Block Second Conv1D Kernel | 1 |
| Style Discriminator Spectrogram-side Block First Conv1D Stride | 1 |
| Style Discriminator Condition-side Block Conv1D Filter | [256, 512, 1024, 1024] |
| Style Discriminator Condition-side Block Conv1D Kernel | 1 |

**Baselines**   We use open source implementation of StarGAN-VC[4], and official implementation of AGAIN-VC[5], AUTOVC[6], and Blow[7]. For fair comparison to other baselines, we does not use pre-trained speaker encoder which is trained with generalized end-to-end (GE2E) loss (Wan et al., 2018). For AUTOVC and VoiceMixer, we train the speaker encoder with the entire model. StarGAN-VC is trained with a batch size of 32 for 200k steps. For AUTOVC, AGAIN-VC, and VoiceMixer, we use mel-spectrogram segment of 192 frames for training. We recommend 192 frames to train the model when the sampling rate is 22,050 Hz in comparison with 128 frames on 16,000 Hz sampling rate. We train AGAIN-VC with the suggested model hyperparameters and a batch size of 64 for 100k steps. We train the AUTOVC with the suggested model hyperparameters and a batch size of 16 for 100k steps. For blow, we train the model with the suggested hyperparameters for 300 epochs over 40 GPU days on Nvidia A100. Other models take under 1 GPU day on Nvidia A100.

For adversarial speaker classification on AUTOVC, we use the same classifier of ours and gradient reversal layer. Then, we use 0.02 weight for adversarial speaker classification loss.

**Mel-spectrogram**   We use the mel-spectrogram as an input for VoiceMixer, AGAIN-VC, and AUTOVC. We use the audio signal downsampled at 22,050 sampling rate. Though a short-time Fouorier transform (STFT) with a window size of 1,024, hop size of 256, and 1,024 points of FFT, We compute Mel-spectrogram. We use 80 channel of mel filterbank spanning 0 Hz to 8 kHz, and clip to a minimum value of $10^{-5}$ before applying log dynamic range compression.

**Vocoder**   For converting mel-spectrogram into audio signal, we use an official implementation of HiFi-GAN [8]. We use the HiFi-GAN generator of $V1$, which has a initial hidden dimension of 512. We train the HiFi-GAN using 108 speakers of VCTK dataset, and we also evaluate the HiFi-GAN as shown in Table 1 and Table 2.

---

[4]https://github.com/liusongxiang/StarGAN-Voice-Conversion

[5]https://github.com/KimythAnly/AGAIN-VC

[6]https://github.com/auspicious3000/autovc

[7]https://github.com/joansj/blow

[8]https://github.com/jik876/hifi-gan

# B   Experiments

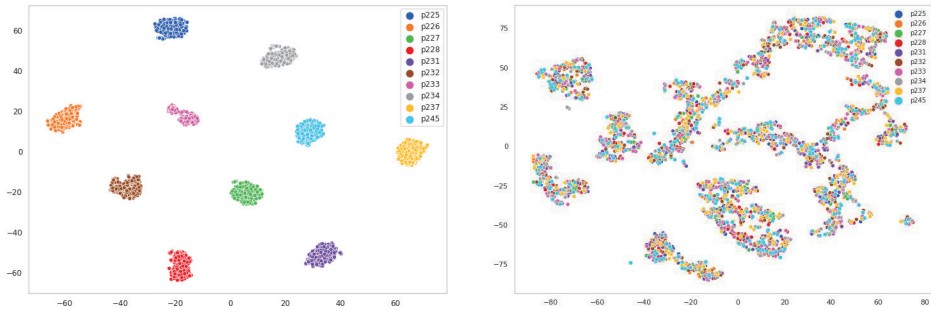

(a) t-SNE visualization for speaker embeddings          (b) t-SNE visualization for content embeddings

Figure 7: t-SNE visualization for speaker/content embedding from different speech of 10 speakers

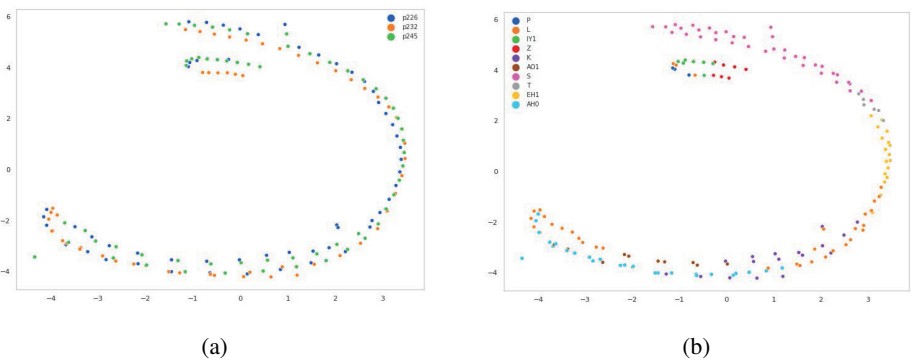

(a)                                                                 (b)

Figure 8: t-SNE visualization for content embeddings of "Please call Stella" from 3 speakers. The left show the content embeddings labeled with speaker id and the right show the same content embeddings labeled with phoneme. The phoneme information is extracted from the attention alignment of Tacotron2, and the character sequence "Please call Stella" becomes [P, L, IY1, Z, , K, A01, L, , S, T, EH1, L, AH0, .].

**t-SNE visualization**    In Figure 7, we present the t-SNE visualization for both speaker and content embeddings from the different utterances of 10 speakers. While the speaker embeddings from different speakers can be distinguished, it is difficult to differentiate between content embeddings from different speakers. This means content encoder extracts the features irrelevant to speaker information. To demonstrate that the content embeddings are related to context information, we also conduct t-SNE visualization for content embeddings as shown in the Figure 8. The content embeddings from the same utterance of different speaker are distinguish by phoneme information. We extract the phoneme label from the attention alginement of Tacotron2, and the character sequence is converted by (Park, 2019).

Table 7: Inference speed comparison

| Model | Latency (s) | Speedup |
|---|---|---|
| AUTOVC | 0.041±0.0199 | - |
| VoiceMixer | 0.007±0.0004 | 5.6× |

**Inference speed**    We compare the inference speed of VoiceMixer compared with the AUTOVC. We conduct the evaluation on a Intel Xeon Gold 6148 CPU and a NVIDIA Titan V GPU with a 1 batch size. For evaluation, both model convert the 400 samples and the average length of mel-spectrogram is 406 frames. Our model has 5.6× speedup compared with the AUTOVC as shown in Table 7.

## C   Evaluation details

**Mean opinion score**   We conduct the subjective MOS test for the naturalness of converted speech and similarity of converted speech to target voice. Figure 9 shows the instructions for participants.

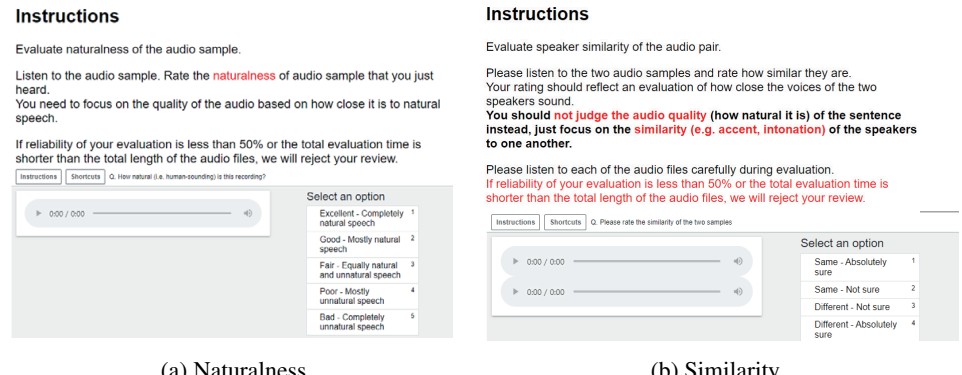

(a) Naturalness                                                     (b) Similarity

Figure 9: Subjective evaluation for Naturalness and Similarity. $0.02 per 1 hit is paid to participants.

**Mean Ceptral Distortion**   We evaluate the model performance with mel cepstral distortion (MCD) (Kubichek, 1993). To compute MCD between synthesized and ground-truth audio, we calculate the first 13 mel-frequency cepstral coefficients (MFCCs) by taking discrete cosine transform to raw waveform. The MCD between two frame is the $l2$ distance between their MFCCs. This can be formulated as follows:

$$MCD_{13} = \frac{1}{T} \sum_{t=0}^{T-1} \sqrt{\sum_{k=1}^{13} (\boldsymbol{M}_{t,k} - \boldsymbol{M}'_{t,k})^2} \tag{22}$$

where $\boldsymbol{M}_{t,k}$, $\boldsymbol{M}'_{t,k}$ represent original and synthesized $k^{th}$ MFCCs of $t^{th}$ frame. $T$ denotes the number of frames. Since two sequences are not aligned, dynamic time warping (DTW) (Berndt and Clifford, 1994) was applied prior to comparison. Here, the lower MCD indicates higher similiarity between two audio.

$F_0$ **Root Mean Square Error**   To evaluate reference similarity in terms of fundamental frequencies ($F_0$), we compute root mean square error for $F_0$ ($RMSE_{F_0}$). We first extract the $F_0$ using open source implementation of World vocoder[9], then computes the $l2$ distance between $F_0$ of synthesized and ground-truth waveform:

$$RMSE_{F0} = 1200 \| (log_2(\boldsymbol{F_r}) - log_2(\boldsymbol{F_s})) \|_2 \tag{23}$$

$\boldsymbol{F_r}$ and $\boldsymbol{F_s}$ represent $F_0$ sequences of raw and synthesized waveform, respectively. We also apply to DTW to calculate $RMSE_{F_0}$ between two sequences, which are not aligned.

**Fréchet Deep Speech Distances**   We report Fréchet Deep Speech Distances (FDSD) (Bińkowski et al., 2020) which assess the quality of generated audio based on the Fréchet distance to the ground-truth audio. FDSD is similar to Fréchet Inception Distance which is the common metric of evaluating GANs for images. However, FDSD is computed on representations extraced from an speech recognition model of DeepSpeech2 Amodei et al. (2016) instead of the Inception network. We follow the open source implementation of FDSD[10], which is computed as follows:

$$FDSD = \sqrt{\| \mu_{\boldsymbol{X}} - \mu_{\boldsymbol{Y}} \|_2^2 + Tr(\textstyle\sum_X + \sum_Y - 2\sqrt{(\sum_X \sum_Y)})} \tag{24}$$

where $\boldsymbol{X}$ and $\boldsymbol{Y}$ refers to extracted feature of ground-truth and generated waveform, respectively. $\mu_{\boldsymbol{X}}$, $\mu_{\boldsymbol{Y}}$ and $\sum_{\boldsymbol{X}}$, $\sum_{\boldsymbol{Y}}$ are the means and covariance matrices of $\boldsymbol{X}$ and $\boldsymbol{Y}$, respectively. $Tr(\cdot)$ denotes the trace of matrix.

---

[9]https://github.com/JeremyCCHsu/Python-Wrapper-for-World-Vocoder
[10]https://github.com/google-research/google-research/tree/master/ged_tts

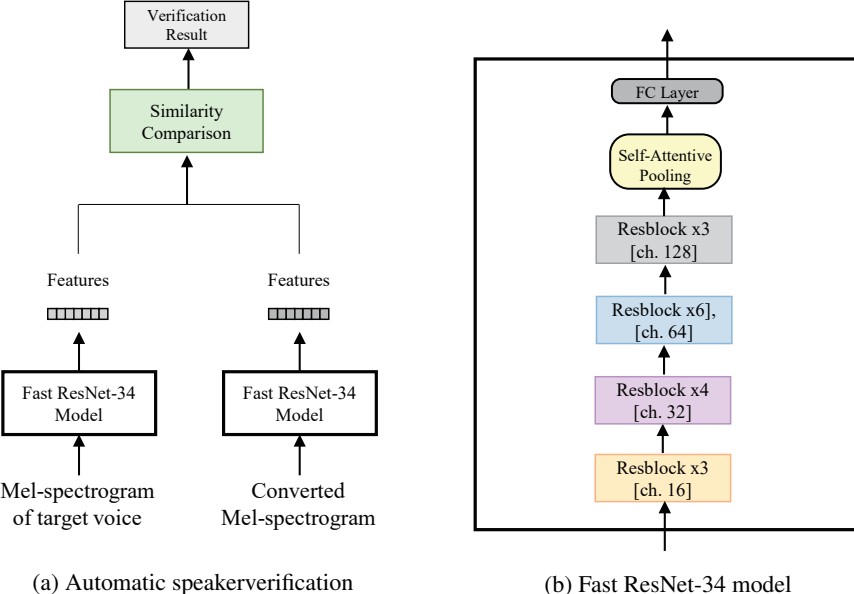

(a) Automatic speakerverification (b) Fast ResNet-34 model

Figure 10: (a) the overall architecture of the automatic speaker verification. (b) the Fast ResNet-34 model.

**Automatic speaker verification**  The automatic speaker verification (ASV) network is shown in Figure 10a. Features are extracted from ground-truth mel-spectrogram of target voice and converted mel-spectrogram by the Fast ResNet-34 model (Chung et al., 2020). Then, the similarity of extracted features are compared to produce final verification result. As shown in Figure 10b, Fast Resnet-34 model is similar to the original Resnet-34 model (He et al., 2016) but contains quarter of the channels. This decrease in the number of channels enables the model to be light-weight and declines computational cost. Compared to the standard Resnet-34 model, the Fast Resnet-34 model drastically reduces the number of parameters from 22 to 1.4 million. Each of the four residual blocks contain [3, 4, 6, 3] layers with [16, 32, 64, 128] filters. The self-attentive pooling (Cai et al., 2018) focuses on informative frames that are crucial for classification task. Finally, the fully-connected layer outputs 512-dimensional feature for similarity comparison.

We use the equal error rate (EER) to calculate speaker verification results. As shown in Figure 11, EER is the location on the ROC curve where the false acceptance rate (FAR) is equal to the false rejection rate (FRR). Lower equal error rate indicates higher accuracy of the ASV system. The Fast ResNet-34 model is trained on the Voxceleb2 (Chung et al., 2018) dataset and tested on the Voxceleb1 (Nagrani et al., 2017) dataset. Compared to other methods, our proposed method achieves lowest ASV EER represented in Table 1.

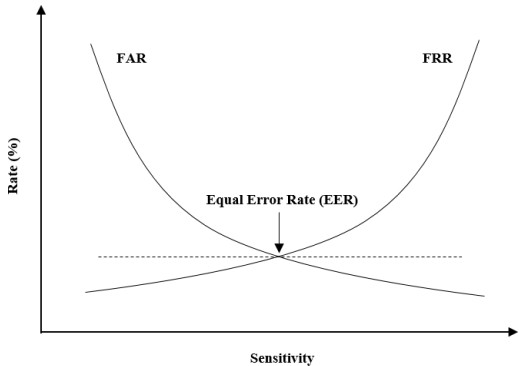

Figure 11: Receiver operating characteristic (ROC) curve

## D   Hyperparameter tuning

We performed hyperparameter tuning during the ablation study. To validate the model, the mel-spectrogram reconstruction loss was used to evaluate the speech quality. However, the lower mel-spectorgram reconstruction loss does not always indicate that the style is effectively transferred in the converted speech. In many cases, the model with too low mel-reconstruction loss value simply reconstructs the source speech and is not able to convert the voice. To validate the voice style transfer performance, we evaluated the converted speech by the speaker classification model during validation. To evaluate the converted speech, a single utterance was selected from each speaker in the test data, and all possible pairs of utterances ($98 \times 98$ = 9,604) were produced. We searched for the weight of loss with a grid search. First, we selected models with a classification accuracy above 95%. Subsequently, we selected the model with the lowest reconstruction loss.

## E   Similarity-based duration

The majority of voice conversion models use the segmented mel-spectrogram as the source speech during training. This aid in causing the data to have the same length in the same batch during training. Without the first and last time frame, the additional time frame $T + 1$ can be used after time $T$. However, in practice, we use the duplicated frame of $T - 1$ as a $T + 1$ frame. As both models are trained almost the same, we use the latter for efficiency. The term $d_n$ is the cumulative sum of the frame number that is added until the similarity is under average similarity. The average similarity is the average over the utterance.