# OpenReview forum: "VoiceMixer: Adversarial Voice Style Mixup"
_NeurIPS.cc/2021/Conference — NeurIPS 2021 Poster_

### Official Review · Reviewer_fx4p · 2021-07-03

**Rating:** 4
**Confidence:** 4

**Summary:**

This paper proposes two ideas to tackle voice conversion task.

First, the authors propose a new information bottleneck approach that aggregates similar content embeddings using similarity-based downsampling&upsampling method.

Second, the authors use adversarial training method to improve the quality of the output. More specifically, the authors propose to use self-learned embeddings to provide conditioning signals to two discriminators, content discriminator and style discriminator.

The proposed VoiceMixer model was compared with previous zero-shot VST baseline models.
Experiment results show that the proposed model shows superior performance than the baseline models.

In Table3, the authors also conduct ablation experiments to see which method is mostly contributing to the superior performance of the proposed model. Experiment results show that the contrastive learning method for content embedding is contributing the most to the performance.

**Limitations And Societal Impact:**

Although the potential negative societal was commented in the Introduction, it was addressed too shortly.
I ask the authors to address this more on Conclusion section and mention what would be the possible solutions to tackle the possible negative impact.

**Main Review:**

Comments on the originality:

This paper proposes several ideas to tackle the VC task.
The quality of the produced speech from this model sounds plausible.

However, I have a few comments on the originality and novelty of this work.

1. Too many methods are used together making the readers hard to follow what is actually contributing to the performance. For example, the authors use Instance Normalization in the content encoder. It is well known that Instance Normalization works as sort of a bottleneck and help remove speaker information from speaker embedding [1]. Therefore, it is unclear how much the proposed information bottleneck approach in 3.4 is actually contributing to the final results in the absence of Instance Normalization. Additionally, the authors proposed to use repulsion feature matching loss, which sounds uncommon to me. Is this really contributing to the final performance at all?
2. More significantly, I have doubt on the originality of this work because very similar ideas were proposed before.
In 3.4, the authors propose to use contrastive learning method and use similarity between adjacent content embeddings to found boundaries. This is actually very similar to the method proposed in [2] where the they also use contrastive learning method and then find the boundary based on similarities between the adjacent embeddings frames.
In addition, the idea of reusing the self-learned embeddings for conditioning discriminator in 3.4 was also proposed previously in [3], where they proposed to learn speaker embedding using contrastive learning and then reuse it to condition the discriminator.

[1] Chou, J. C., Yeh, C. C., & Lee, H. Y. (2019). One-shot voice conversion by separating speaker and content representations with instance normalization. Interspeech 2019.

[2] Kreuk, F., Keshet, J., & Adi, Y. (2020). Self-supervised contrastive learning for unsupervised phoneme segmentation. Proc. Interspeech 2020.

[3] Choi, H. S., Park, C., & Lee, K. (2020). From inference to generation: End-to-end fully self-supervised generation of human face from speech. Proc. ICLR 2020.



Comments on writings:

- Pros
1. The proposed method is easy to follow.

- Cons
1. It seems like the authors tried to address the negative societal impact by mentioning in the Introduction section very shortly as written below.

17 VC is also called voice style transfer (VST),
18 and it shares a long history with the objective to clone someone’s voice. There is even a potential
19 risk of usage in crime such a voice spoofing (Kinnunen et al., 2012),

 This deteriorates the flow and logical structure of the writing. I recommend the authors to address this issue in the Conclusion section for better Introduction.



Some other comments & questions:

1. Why is Similarity(%) results in zero-shot VST experiment better than that of many-to-many experiment? Maybe there was something wrong in the experiment setting?
2. Is it okay to conduct human evaluation without IRB? (as written in checklist)
3. The authors argue that the alignment found by the proposed method is actually related to "phoneme alignment". However, I'm not sure if the proposed method is actually related to "phoneme-alignment". I don't think Fig4. (c) and Fig4. (d) are similar. In addition, I don't think simply showing figures (in Fig.4) support the argument.

**Time Spent Reviewing:**

4 hours

---

> ### Author Response · Authors · 2021-08-10
> **Responses to Reviewer fx4p**
>
> We appreciate for your helpful comments and suggestions. We have provided responses to your questions below to address your concerns.
>
> [About Instance normalization]
> Although instance normalization is often used to remove the speaker information from the source speech in the voice conversion domain, the use of instance normalization alone is not sufficient to remove the speaker information. For example, AGAIN-VC also uses instance normalization, but the content embedding includes speaker information as included in Table 4. All models of our ablation study also used the instance normalization, however, our final model shows better performance in the voice style transfer.
>
> [About repulsive style loss]
> It measures the feature difference between each block of discriminators. The feature matching loss is a learned similarity metric between the features. The repulsive style loss measures the style-conditioned feature difference between each block of the discriminator from the source and converted speech. Maximizing the style feature distance between the source and converted speech prevent the converted speech from containing the style of the source speech.
>
> [About the contrastive learning]
> In recent years, the contrastive learning method has been adopted to extract important representations in speech tasks. In our work, we modified the method to disentangle the content and style information from the source speech.
>
> [About the similarity results]
> In most speech domains, human evaluation is conducted through crowdsourcing, such as Amazon Mturk. As each evaluation (many-to-many VST, zero-shot VST) is conducted independently, the subjectivity of the evaluation may differ for each evaluation. To address this issue, we conducted a subject evaluation with two anchors (ground-truth and Vocoded audio). For example, the results of [1] also have better performance in the zero-shot VST experiment than many-to-many VST, but it is okay.
>
> [About the IRB]
> The lack of IRB in our human evaluation was not problematic. We did not observe human activity and the study was not conducted while facing the subjects for research as we used the crowdsourcing service.
>
> [About the alignment]
> The alignment is related to the content information. As we make the alignment according to the similarity of the content information, the alignment may be related to the content features, not only phonetic information. It is important to guide similar content to have similar embedding. For example, the model that was trained without contrastive loss shows almost diagonal alignment, which implies that content and style information are not disentangled effectively.
>
> [About the ethical concerns]
> We will add more discussion about potential negative societal impact of voice conversion.
>
> [1] S. Yuan et al., "Improving Zero-Shot Voice Style Transfer via Disentangled Representation Learning," International Conference on Learning Representations, 2020.

---

> > ### Comment · Reviewer_fx4p · 2021-09-15
> > **Final brief response**
> >
> > Although some of the concerns have been addressed, I'm still not sure whether the similarity-based information bottleneck alone can act as a solid information bottleneck method without the compounding proposed methods.Therefore, I keep my initial score.

---

### Official Review · Reviewer_Yn2G · 2021-07-15

**Rating:** 7
**Confidence:** 5

**Summary:**

This paper proposes a new information bottleneck design to disentangle speaker style information from the content information in speech for voice conversion. The information bottleneck downsamples the hidden representation based on representation similarity (instead of uniform downsampling as in the AutoVC baseline). The proposed voice conversion system also integrates some auxiliary loss to further improve the performance.

**Limitations And Societal Impact:**

The discussion about limitations and negative societal impact is adequate.

**Main Review:**

Pros of this paper:

1. The idea of similarity-based bottleneck is novel and intuitively plausible.
2. This paper evaluates the performance of voice conversion systems through very comprehensive objective and subjective metrics.
3. The audio demo sounds very compelling. It is among the highest-quality demos for non-parallel unsupervised voice conversion systems. The demos for the baselines also sound pretty nice.

Cons of this paper:

1. On top of the proposed novel information bottleneck design, this paper introduces a lot of loss terms, most of which are like simple superpositions of known techniques. For the generator, there are four losses during reconstruction, and four losses during conversion. It is the most complicated voice conversion system I have seen so far.

While the complication can be partially justified by the good performance, there are some outstanding issues. First, this paper does not discuss in detail how the hyper-parameter tuning is performed. Considering the number of loss terms, tuning the loss weights looks like a big challenge. Second, the ablation study was not performed thoroughly. In particular, the main novelty in this paper is an improved bottleneck design that disentangles speaker style information, but a lot of the loss terms also aim to disentangle speaker info, Ladv, Lpos, Lneg, L*adv, L*style+, and L*style-. It is thus unclear how much the improved bottleneck design plays a role. In Table 3, some ablation studies are performed, but a more thorough ablation study would be desirable, considering the large number of loss terms. For example, I would be interested in seeing an AutoVC baseline with only the bottleneck replaced with the similarity-based one.

2. On a related note, as the major innovation of this paper, the information bottleneck design is not fully studied by the experiment. For example, the paper claims that similarity-based IB frees the need to tune the hidden representation sampling rate, and thus achieves a better trade-off between speaker style disentanglement and content loss. I would love to see this point better validated. For example, one possible experiment is to plot the disentanglement and content loss (choose one of the many objective metrics in the paper for each) against the physical bottleneck dimension. If the authors' claim holds, AutoVC should exhibit a very clear trade-off as bottleneck dimension changes, but the proposed algorithm should not be as sensitive.

Minor issues:
1. The authors seem to disambiguate between the adversarial loss for the discriminator and the generator using only the argument difference, i.e. Ladv(Dc, Ds; G) v.s. Ladv(G; Dc, Ds). This is not quite a standard notation. Try L^D_{adv}(Dc, Ds; G) v.s. L^G_{adv}(G; Dc, Ds).

2. Recently, [1] proposed a similar similarity-based information bottleneck. The general goals of the IB design are different in these two papers (one is for disentangling rhythm, and this work is for better disentangling speaker style), but the practice is similar. It is worthwhile to mention this work in the related work section.

[1] Qian, Kaizhi, et al. "Global Prosody Style Transfer Without Text Transcriptions." International Conference on Machine Learning. PMLR, 2021.

**Time Spent Reviewing:**

5 hours

---

> ### Author Response · Authors · 2021-08-10
> **Responses to Reviewer Yn2G**
>
> We appreciate for your helpful comments and suggestions. We have provided responses to your questions below to address your concerns.
>
> [About the hyperparameter tuning]
> We performed hyperparameter tuning during the ablation study. To validate the model, the mel-spectrogram reconstruction loss was used to evaluate the speech quality. However, the lower mel-spectorgram reconstruction loss does not always indicate that the style is effectively transferred in the converted speech. In many cases, the model with too low mel-reconstruction loss value simply reconstructs the source speech and is not able to convert the voice. To validate the voice style transfer performance, we evaluated the converted speech by the speaker classification model during validation. To evaluate the converted speech, a single utterance was selected from each speaker in the test data, and all possible pairs of utterances (98 x 98 = 9,604) were produced. We searched for the weight of loss with a grid search. First, we selected models with a classification accuracy above 95%. Subsequently, we selected the model with the lowest reconstruction loss.
>
> [About the bottleneck experiment]
> When we trained the fixed-length information bottleneck version of our model without auxiliary losses (L_advsc, L_pos, and L_neg), these models ($\tau$ =16 and $\tau$ = 32) only reconstruct the source speech. We think that because our model has a large channel dimension, the fixed-length information bottleneck cannot disentangle the content and style information. We also trained the fixed-length information bottleneck version of our model with auxiliary losses, as indicated in Table 3. The results demonstrate that the similarity-based information bottleneck exhibited higher performance than the fixed-length information bottleneck.
>
> [About the adversarial loss notation]
> Thanks for your advice. We will fix the notation in the revised version.
>
> [About adding the related work]
> Thanks for your suggestion. We also think this paper [1] proposed a similar method. As you mentioned, it is interesting to convert the rhythm of speech. We will add it in the revised version.
>
> [1] K. Qian, et al, "Global Prosody Style Transfer Without Text Transcriptions." International Conference on Machine Learning, 2021.

---

> > ### Comment · Reviewer_Yn2G · 2021-08-20
> > **Major concerns addressed**
> >
> > I would like to thank the authors for their response. I encourage the authors to add the details of their hyperparameter tuning to the paper. I am raising my score from 6 to 7.

---

### Official Review · Reviewer_NNWU · 2021-07-15

**Rating:** 8
**Confidence:** 4

**Summary:**

This work expands on recent work in autoencoder-based voice conversion models by using a novel variable-length similarity-based information bottleneck trained with contrastive loss inspired by self-supervised representation learning, and avoids complications around choosing a fixed downsampling ratio. In addition, it introduces a pair of adversarial discriminators for better disentangling of content versus speaker. It can be trained without any paired data or additional information such as text transcriptions or pitch contours, and achieves significantly higher objective and subjective scores for naturalness and speaker similarity compared to previous work for both many-to-many and zero-shot voice style transfer.

**Ethical Concerns:**

2. Raise safety or security concerns. For example: is there a risk that applications could cause serious accidents or open security vulnerabilities when deployed in real-world environments?

Voice conversion can exploit speaker id based security setups.

7. Deceive people in ways that cause harm. For example: could the approach be used to facilitate deceptive interactions that would cause harms such as theft, fraud, or harassment? Could it be used to impersonate public figures to influence political processes, or as a tool of hate speech or abuse?

Voice conversion can exacerbate the issue of fake news.


The ethical concerns have not been addressed except for a single line about the potential for voice spoofing in the introduction. However, this work is building on previous work, and improves the speaker verification equal-error-rate metric from 14.0% (AUTOVC (τ=32)) to 12.5%, which is likely not a significant enough improvement to unlock any new avenues of attack. The improvement in Naturalness for the zero-shot VST may be a cause for concern. I have marked this as needs ethics review purely based on the quality of the zero-shot VST in the audio samples, which sound alarmingly real.

**Ethics Review Area:**

["Inappropriate Potential Applications & Impact  (e.g., human rights concerns)"]

**Limitations And Societal Impact:**

The submission acknowledges that some loss of content information is unavoidable with their approach. The submission acknowledges that voice style transfer can be used maliciously. However, the submission does not propose any mitigation strategies.

**Main Review:**

# Originality
This work builds on ideas from related work and combines them in a novel way for the target task. As far as I am aware, the variable-length similarity-based information bottleneck is novel, and the combination of autoencoder and disentangled GAN is likely to be novel as well.

# Quality
The submission is high quality with multiple baselines and relevant metrics both subjective and objective. Audio samples and code are included, as well as a detailed description of model architecture and hyperparameters.

The claim that the proposed model results in only a small loss of content information (with the implied claim that baseline models have a larger loss of content information) could be better supported with comparing ASR metrics, as the groundtruth transcripts are available for VCTK.

I personally would have also preferred some more (subjective) analysis on other aspects of "style" beyond speaker identity, for example pacing, pause lengths, and accents.

# Clarity

There are parts of the submission that are not very clear, but overall it is understandable.

# Significance

I believe that the proposed variable-length similarity-based content autoencoder is a significant contribution, not just for VST, but potentially for other speech tasks such as TTS or ASR. The ability to obtain phoneme durations from just raw audio can be important. It would be interesting to see if this approach can be improved when text transcriptions are available, and whether the result can be more accurate than alignments from FastSpeech2.

# Misc

* Section 1 Introduction
  * line 19 "such a voice spoofing"
  * line 56 "Self-supervised learned similarity makes the information bottleneck disentangle the content and style without effort to find the proper down sampling size."
    * This line is a bit confusing to read. Can you try to rephrase it?
  * Consider adding a reference to _Jia, Ye, et al. "Transfer Learning from Speaker Verification to Multispeaker Text-To-Speech Synthesis." Advances in Neural Information Processing Systems 31 (2018)._
* Section 2 Background
  * personally I find it a bit distracting that $f_c, f_s$ are given in (content, style) order yet $X_{1,A}$ is given in (style, content) order.
  * line 85/86 "too narrow"/"too wide"
    * Can you confirm my understanding here? A narrow bottleneck means a large value for $\tau$ and fewer downsampled feature frames. I would have thought that this would lead to lower reconstruction quality.
  * "In the process of separating content and style information, some content information is lost even with proper bottleneck size. Therefore, missing some content information in converted voice is inevitable."
    * I'm not sure I agree with this conclusion theoretically.
* Section 3 VoiceMixer
  * Section 3.2 Similarity-based information bottleneck
    * What is the value of $\mathbf{c}_{T+1}$?
    * "where $\mathbf{d}_n$ is cumulative sum until the similarity $q_t$ is under the average similarity"
      * This part is not very clear. Can you clarify it?
  * Section 3.3 Auxiliary losses for similarity
    * What is $k$? Is it a constant? If it is a constant then no sampling is happening. What about content embeddings near the end of the sequence?
  * Figure 3
    * "Recontructed Speech"
  * Section 4.6 Content and speaker disentanglement
    * line 296/297 "speaker identify"
* Appendix A
  * Figure 5
    * "SoftPuls"
  * Figure 6
    * "Discrminator"
  * Table 5
    * "Spaker"
* Audio demos
  * The audios starting from Zero-shot VST do not load

**Needs Ethics Review:**

Yes

**Time Spent Reviewing:**

5

---

> ### Author Response · Authors · 2021-08-10
> **Responses to Reviewer NNWU**
>
> We appreciate for your helpful comments and suggestions. We have provided responses to your questions below to address your concerns.
>
> [About the WER/CER results]
> Thanks for your advice. We conducted the automatic speech recognition (ASR) test for converted speech. The results show VoiceMixer revealed a lower word error rate (WER) of 20.92% (GT: 8.18%, Vocoder: 10.12% AutoVC ($\tau$ = 16): 39.27%, AutoVC ($\tau$ = 32): 61.99%) and VoiceMixer also revealed a lower character error rate (CER) of 22.13% (GT: 8.612%, Vocoder: 10.19%, AutoVC ($\tau$ = 16): 41.52%, AutoVC ($\tau$ = 32): 65.40%. We used the Google Speech-to-Text API for ASR model. We will add this result in the revised version.
>
> [About the extension to other speech tasks]
> Thank you for your suggestion to apply this method to other speech tasks, such as TTS or ASR. Our initial aim was to develop a voice conversion model using similarity-based information bottleneck without text transcripts. However, we are planning to extend our method to train the TTS model without text transcripts. For example, the TTS model can be trained with dataset 1 (audio with a text transcript) and dataset 2 (audio only). When the model is contructed with two encoders and a shared decoder, the first encoder synthesizes the speech from the text, and the second encoder synthesizes the speech from the audio (as in our model). In this manner, the model can synthesize speech using dataset2 from the text.
>
> [About adding reference]
> Thank you for your advice. We will add the paper [1] in Section 1. Moreover, we also think that this paper is a starting point for transferring the unseen speaker style in the TTS domain; therefore, it is strongly related to ours.
>
> [About the notation]
> Thanks for your advice. We change the order as (style, content).
>
> [About the “too narrow”/”too wide” information bottleneck]
> We aim to convey that a ‘’too narrow bottleneck size’’ is a small value for \tau and ‘’too wide bottleneck size’’ is a large value for $\tau$. To avoid misunderstanding in relation to the original AutoVC paper, we will clarify this in the revised version.
>
> [About the typo]
> Thanks for your comments. We will fix the typo in the revised version.
>
> [About the similarity-based duration]
> The majority of voice conversion models use the segmented mel-spectrogram as the source speech during training. This aid in causing the data to have the same length in the same batch during training. Without the first and last time frame, the additional time frame T+1 can be used after time T. However, in practice, we use the duplicated frame of T-1 as a T+1 frame. As both models are trained almost the same, we use the latter for efficiency.
> The term $d_n$ is the cumulative sum of the frame number that is added until the similarity $q_t$ is under average similarity. The average similarity is the average over the utterance. We will clarify this part in the revised version.
>
> [1] Y. Jia, et al., "Transfer Learning from Speaker Verification to Multispeaker Text-To-Speech Synthesis," Advances in Neural Information Processing Systems, 2018.
>
> [About the equation 7]
> We set k to 24 (about 0.3s) which is over the average duration of consonant-vowel syllables. Thank you for pointing out the case in which no sampling occurred. There is a mistake in Equation 7. We only use the time frame from k to T-k to calculate the contrastive loss for negative samples in practice. We will fix the equation in the revised paper.
>
> [About the audio demos]
> We checked our demo page, and it appears to be working satisfactorily in our environment. However, for your convenience, we will include the download link for the audio demo files in the revised paper. We highly recommend that you listen to our audio demo sample!
>
> [About the ethical concerns]
> We acknowledge the potential risks the Naturalness improvement of the zero-shot VST might cause in real-world environments. We will make sure to build upon the discussion of potential negative impact on speaker ID based security setups or the issue of fake news, as you mentioned.

---

### Official Review · Reviewer_H8yZ · 2021-07-19

**Rating:** 8
**Confidence:** 4

**Summary:**

The paper presents VoiceMixer for voice conversion, which can decompose the content and voice style in a self-supervised manner.  More specifically, the paper introduces a self-supervised representation learning method to extract the content embedding and the voice style embedding, without the necessity to condition on external features (such as additional text transcriptions).  Furthermore, the paper also proposes an adversarial voice style mixup method which contains two separate discriminators for disentangled content and style representation learning respectively.

Compared with existing self-supervised method like AutoVC, one advantage of the proposed method is there is no need to find the proper size of the information bottleneck (e.g. down-sampling in channel and temporal, from Auto VC) heuristically.


**Limitations And Societal Impact:**

(1) Line 283-285: it is said “…when trained without any information bottleneck or all of the auxiliary losses, these models are not able to covert any voice, but only reconstruct source speech”.  Would the authors please specify which line of Table 3 corresponds to this conclusion?

(2) Line 9/64/275/307/314: the authors have mentioned several times “… with only a small loss of content information”.  Would the authors please clarify this conclusion by adding some experimental results to support this and also giving more explanation about this why there is such a small loss?

(3) In the work of AUTOVC (Qian et al., 2019), it is said that “The down-sampling can be regarded as dimension reduction along the temporal axis, which, together with the dimension reduction along the channel axis, constructs the information bottleneck.”  The downsampling operation should be conducted both along temporal axis and channel axis.  However, it seems the proposed “similarity-based information bottleneck” by the paper only considers the downsampling along the temporal axis.  The authors should either explain why no downsampling along channel axis is not considered, or add more experiments for downsampling along channel axis.


**Main Review:**

[Originality]

(1) The paper poses a similarity-based information bottleneck with self-supervised representation learning framework, in which the content embeddings are downsampled according to the cosine similarity between adjacent phonetic content embeddings.  Furthermore, to decide the dynamic bottleneck size, Gaussian downsampling is adopt with reference to Shen et al. 2020. (Line 116-117) by assuming that the center of same content has the largest information.

(2) To increase the similarity between adjacent content embeddings, the paper also introduces contrastive loss from wav2vec (Schneider et al, 2020) as well as the adversarial speaker classification.  The method sounds technically reasonable.

Schneider et al. 2020. Wav2vec: Unsupervised Pre-training for Speech Recognition.

(3) Different from Cycle-GAN, the paper proposed separate content and style discriminators for adversarial feedback by following the autoencoder based reconstruction for model training.

[Quality and Clarity]

Most of the claims are well supported.  More importantly, in Section 3, the paper has been written in a clear way, including the similarity-based information bottleneck, the use of several different losses including the GAN losses.

Detailed structures of the proposed model as well as the code has been given in the supplementary materials.  It would be easy for other researchers of the community to follow the work.

For some detailed comments, please find it in the following “Limitations” part.


**Time Spent Reviewing:**

10 hours

---

> ### Author Response · Authors · 2021-08-10
> **Responses to Reviewer H8yZ**
>
> We appreciate for your helpful comments and suggestions. We have provided responses to your questions below to address your concerns.
>
> [About the line 283-285]
> We trained two additional models. The first is the model that was trained without an information bottleneck. In this model, the encoder output is directly fed to the decoder without any information bottleneck. The other is the model that was trained without positive contrastive loss, negative contrastive loss, and adversarial speaker classification loss. Both models only reconstructed the source speech and were unable to convert any voice. Therefore, a subjective evaluation was not necessary. Hence, we have only briefly presented the results of these two models in the manuscript, and not in Table 3.
>
> [About the small loss of content information]
> To evaluate the loss of content information, we conducted the FDSD evaluation as shown in Table 1. To support our statement, we additionally conducted word error rate (WER) and character error rate (CER) evaluation. The results show VoiceMixer has a lower WER of 20.92% (GT: 8.18%, Vocoder: 10.12% AutoVC ($\tau$ = 16): 39.27%, AutoVC ($\tau$ = 32): 61.99%) and VoiceMixer also has a lower CER of 22.13% (GT: 8.612%, Vocoder: 10.19%, AutoVC ($\tau$ = 16): 41.52%, AutoVC ($\tau$ = 32): 65.40%. We used the Google Speech-to-Text API for ASR model. We will add this result in the revised version.
>
> [About the information bottleneck along the channel axis]
> In AutoVC, dimension reduction should be performed along the channel axis to disentangle the content and style information. When training the AutoVC with a channel size of 256 (128 in each direction), this AutoVC model only reconstructs the source speech, but the reconstructed speech of this model has a lower reconstruction loss than the AutoVC model with the original channel size of 64 (32 in each direction). In this regard, we believe that a trade-off exists between the speech quality and conversion performance in the channel dimension reduction. Although our model has a larger channel dimension size and does not use dimension reduction along the channel axis, it can disentangle the content and speaker information as indicated in Table 4. Hence, our model exhibits higher performance in terms of naturalness without the need for a time-consuming search for the proper bottleneck size.

---

> > ### Comment · Reviewer_H8yZ · 2021-09-04
> > **Concerns addressed**
> >
> > I appreciate the responses from the author.  They have addressed my concerns especially for the small loss of the content.

---

### Review · Ethics_Reviewer_tLu4 · 2021-08-08

**Recommendation:**

I do believe that the authors can address the potential negative societal impacts (and potential ways to mitigate these risks). This might require building out a new, brief section at the end in a consolidated fashion to better surface the potential negative societal impacts. In such a section, the authors could (and should, per the guidelines) include a discussion of several potential mitigation approaches.

As just two examples of (imperfect) potential mitigation approaches:

1. Deployment could be conditioned on *written consent from all parties involved* (e.g., target voice and source speech). Note: even this may have potential issues, though, given the difficulties of meaningful consent when one party is dead (the deceased's estate is one plausible avenue, but even that maybe insufficient and/or too narrowly tailored).

2. Deployment could be conditioned on *mandatory disclosure that the final content/media is synthetic in nature.* Note: what form the disclosure comes in can take a variety of forms that are plausibly acceptable.

Overall, I would encourage the authors to dedicate a specific discussion section to the potential negative societal impacts, rather than merely include it in the conclusion. I believe the potential negative societal impacts, as well as potential mitigation approaches and policies, are robust enough to merit their own dedicated section.

**Ethical Issues:**

Yes

**Ethics Review:**

This paper, which builds on related work in voice conversion, inherently raises ethical concerns. While the specific methodology and novel contributions may not uniquely raise completely new ethical concerns, they should nonetheless be addressed.

According to the NeurIPS ethical guidelines, submissions are to include a discussion about the potential negative societal impacts of the proposed research artifact or application. I believe this paper does an insufficient job of doing so. The authors indicate that they included potential negative societal impacts of the work in Section 1, but Section 1 only includes a single sentence on potential negative consequences:

>  There is even a potential risk of usage in crime such a voice spoofing (Kinnunen et al., 2012), and also in various applications in entertainment (Nachmani and Wolf, 2019), education (Sisman et al., 2020), security (Wu and Li, 2016), and voice restoring (Yamagishi et al., 2012).

While all of this is true, I believe this is far too limited a discussion of the potential negative impacts. Specifically, voice conversion can also raise the following ethical issues:

1. Voice conversion can raise *safety or security concerns.*

For example, such technology could be used to exploit and/or spoof speech-based security/biometric systems. I'm personally unfamiliar with such systems and their design and counter-measures, but one can easily imagine it.

2. Voice conversion can *deceive people in ways that contribute to or directly harm.*

I believe that the potential political harms of such technology are noted publicly (though not in this paper) and do not need significant elaboration in this review. Voice conversion/VST could clearly be used in a malicious way to impersonate prominent public figures in ways that would directly influence political processes.

Just as important, this technology could also facilitate deceptive interactions that could lead to fraud, theft, abuse, or harassment. In particular, such technology could fuel intimate partner violence (which can range from abusive acts to harassment), could exacerbate fraud schemes (especially elder fraud), could exacerbate online harassment campaigns, and provide tools for white supremacists to racially abuse and harass individuals. Each of these potential harms could also constitute human rights concerns or could constitute a detrimental effect on an individual's livelihood or economic security, as well.

According to the NeurIPS ethical guidelines, whenever potential negative social impacts are identified within a paper, "submissions should also include a discussion about how these risks can be mitigated." Based on my reading, this paper does not include discussion of potential mitigation approaches.

I also believe that the paper could briefly elaborate on the very real positive effects of this technology, when deployed safely and securely. In particular, the paper could surface some benefits to individuals with vocal disabilities.

---

### Review · Ethics_Reviewer_oFLP · 2021-08-12

**Recommendation:** See above.

**Ethics Review:**

The paper presents a new voice mixing approach that decompose and transfer voice styles in a self supervised manner. While the paper posts significant improvement in performance, and is built on prior work, the presented research in and of itself doesn't introduce or exacerbate ethical challenges. However the paper could add a more detailed discussion on the limitations, especially around content loss and potential scenarios where that might cause harms to people, especially if it might impact certain groups of people worse than others. Similarly, since malicious use cases of this technology around deception and fraud are a problem, it would be great for the authors to discuss a bit more about these issues in the paper. In particular, if there are malicious uses you can't mitigate, it would be great to discuss this.

---

### Author Response · Authors · 2021-08-22
**Response to Ethics Reviewers**

Thank you for your sincere concerns and advice about the ethical issues. We all acknowledge the potential harms, malicious use, and ethical concerns of voice conversion technology. We will promise to elaborate on the discussions of these risks and mitigation strategies in the revised version. As you suggested, we will add a section for the potential negative societal impacts. We acknowledge that we should have stated the potential negative impact and the mitigation strategies in more thorough manner. We will discuss the social negative impacts on voice conversion and the mitigation strategies in the revised version, as below.

Recently, as speech-related AI systems such as voice conversion (VC) and text-to-speech (TTS) synthesize realistic audio, there is an increased risk of the potential harms, malicious use, and ethical concerns for TTS and VC systems. These systems can be used to deceive people in various ways; a usage in crime such a voice spoofing [1], fake news [2], and commercial use by cloning voice without consent [3]. It is necessary to consider a countermeasure for reducing these potential risks. The anti-spoofing techniques [4] such a fake audio detection [5-7] are developed to distinguish synthesized speech, and [8] demonstrated that the AI outperforms the human in fake audio attacks. Moreover, recently, there is an attack using a partially fake audio clip, where some small fake audio clips are hidden in real speech audio. As [9] indicated that partially fake audio detection is more challenging than fully fake audio, it is also important to develop a partially fake audio detection model. The explainable AI (XAI) techniques [10, 11] are adopted to detect the deepfake image and to visualize their representations for the explainability of deepfake detection [12], and it can be extended to an explainable model for fake audio detection.

Additionally, as mentioned by the ethics reviewers, we will open a discussion section in the revised version of the paper to discuss potential ways to mitigate risks involved with our method. As suggested, we agree that discussions of detailed potential mitigation approaches for our research is a step in the right direction. Our approach on the usage of self-supervised method for voice style transfer coincides with the suggestion that both source and target voice parties must give written consent. Furthermore, if a party is dead, we either believe that this voice conversion technology should not be used or used conditionally with the consent from a family member. However, there is much to be discussed on this matter in detail, which we will re-emphasize that a whole new section will be dedicated for.

[1] Kinnunen, Tomi, et al. "Vulnerability of speaker verification systems against voice conversion spoofing attacks: The case of telephone speech." 2012 IEEE International Conference on Acoustics, Speech and Signal Processing (ICASSP). IEEE, 2012.

[2] Singhal, Shivangi, et al. "Spotfake: A multi-modal framework for fake news detection." 2019 IEEE fifth international conference on multimedia big data (BigMM). IEEE, 2019.

[3] Wang, Run, et al. "Deepsonar: Towards effective and robust detection of ai-synthesized fake voices." Proceedings of the 28th ACM International Conference on Multimedia. 2020.

[4] Wu, Zhizheng, et al. "Detecting converted speech and natural speech for anti-spoofing attack in speaker recognition." Thirteenth Annual Conference of the International Speech Communication Association. 2012.

[5] Tak, Hemlata, et al. "End-to-end anti-spoofing with RawNet2." ICASSP 2021-2021 IEEE International Conference on Acoustics, Speech and Signal Processing (ICASSP). IEEE, 2021.

[6] Mittal, Trisha, et al. "Emotions Don't Lie: An Audio-Visual Deepfake Detection Method using Affective Cues." Proceedings of the 28th ACM international conference on multimedia. 2020.

[7] Ma, Youxuan, et al. "RW-Resnet: A Novel Speech Anti-Spoofing Model Using Raw Waveform." arXiv preprint arXiv:2108.05684 (2021).

[8] Müller, Nicolas M., Karla Markert, and Konstantin Böttinger. "Human perception of audio deepfakes." arXiv preprint arXiv:2107.09667 (2021).

[9] Yi, Jiangyan, et al. "Half-Truth: A Partially Fake Audio Detection Dataset." arXiv preprint arXiv:2104.03617 (2021).

[10] Gunning, David, et al. "XAI—Explainable artificial intelligence." Science Robotics 4.37 (2019).

[11] Bach, Sebastian, et al. "On pixel-wise explanations for non-linear classifier decisions by layer-wise relevance propagation." PloS one 10.7 (2015): e0130140.

[12] Malolan, Badhrinarayan, Ankit Parekh, and Faruk Kazi. "Explainable deep-fake detection using visual interpretability methods." 2020 3rd International Conference on Information and Computer Technologies (ICICT). IEEE, 2020.

---

> ### Comment · Ethics_Reviewer_tLu4 · 2021-08-28
> **Brief response**
>
> I believe the proposed approach by the authors addresses my concerns, and I appreciate the authors' response in discussion. I also think that this expanded discussion would comport with and not conflict with the review of the other ethics review, but do not want to preempt any discussion by that reviewer.
>
> Beyond some of the cited papers, I would encourage the authors' in their discussion to approach some of the topics mentioned in my original comment, given that I think they are rather foreseeable potential harms (e.g., I might encourage the authors to disambiguate "crimes," and give mention to certain behaviors that, criminalized or not in certain countries, are nevertheless harmful — such as IPV and harassment).
>
> Thanks again to the authors for their response.

---

### Decision · Program_Chairs · 2021-09-28

**Decision:**

Accept (Poster)

**Comment:**

UPDATE: The revision was reviewed and paper accepted.

----

This paper was discussed at length between the SACs, ACs, ethics reviewers, ethics review chairs, and program chairs.  In the end, a decision was made to conditionally accept the paper.  The list of conditions for acceptance is as follows:

1. Meaningful broader impacts statement. Moving beyond superficial discussion of obvious harms into a more detailed and thorough reflection on ethical issues, especially possible broader impact and potential for misuse.

2. Restricted release of model through some type of licensing or form-restricted access (ie. private repo accessed via request, model code and data use restricted by licenses, etc.)

3. Discussion of possible theoretical and practical mitigation strategies for minimizing harm of such technologies in the future. If this is not possible to discuss, include a clear articulation of the limits of such models in the absence of mitigation approaches. It is not necessary to implement the mitigation strategies discussed though some current work in this area should be highlighted by authors in the main text.

The original meta-review from the AC follows.

---

This paper proposes a similarity-based information bottleneck with self-supervised representation learning and adversarial training to decompose the content and style for voice conversion. Reviewers have many discussion on the pros and cons of this work. Although some of the techniques used in the paper have been proposed before, the integration of these techniques is non-trivial and novel. Some reviewers have concerns on the experiment results, and suggest the authors could add more detailed ablation studies to verify the effectiveness of each single component in the method, which is expected to be done in the camera ready paper if being accepted. Ethics reviewer also had some concerns on ethical issues but the authors have explained that well. Overall, it is a good paper and I recommend to accept as a poster.

**Consistency Experiment:**

NeurIPS has a long history of experimentation. In 2014, NeurIPS ran an experiment in which 10% of submissions were reviewed by two independent committees to quantify the randomness in the review process. This year, we repeated a variant of this experiment to see how the quality of the review process has changed over time.  This paper was part of the experiment and was therefore assigned to two committees (consisting of reviewers, an Area Chair, and a Senior Area Chair) that reached independent decisions.  If both committees made the same recommendation, this recommendation was followed. If a single committee recommended acceptance, the paper was accepted (with the exception of a few cases in which the other committee identified what we considered a fatal flaw, e.g., an error in a key result).

Both committees reached the same decision: **Accept (Poster)**

The other committee assigned to the paper recommended **Accept (Poster)**.  You can find the other set of reviews, along with any follow up discussion with the authors here:
https://openreview.net/forum?id=1Sy9EwFCyFQ